# ONLINE MINIMIZATION OF POLARIZATION AND DISAGREEMENT VIA LOW-RANK MATRIX BANDITS

**Federico Cinus**[1]**, Yuko Kuroki**[1]**, Atsushi Miyauchi**[2]**, Francesco Bonchi**[1]

[1]Intesa Sanpaolo AI Research    [2]Intesa Sanpaolo

{federico.cinus, yuko.miyauchi, atsushi.miyauchi, francesco.bonchi} @intesasanpaolo.com

## ABSTRACT

We study the problem of minimizing polarization and disagreement in the Friedkin–Johnsen opinion dynamics model under incomplete information. Unlike prior work that assumes a static setting with full knowledge of agents' innate opinions, we address the more realistic online setting where innate opinions are unknown and must be learned through sequential observations. This novel setting, which naturally mirrors periodic interventions on social media platforms, is formulated as a regret minimization problem, establishing a key connection between algorithmic interventions on social media platforms and the theory of multi-armed bandits. In our formulation, a learner observes only a scalar feedback of the overall polarization and disagreement after an intervention. For this novel bandit problem, we propose a two-stage algorithm based on low-rank matrix bandits. The algorithm first performs subspace estimation to identify an underlying low-dimensional structure, and then employs a linear bandit algorithm within the compact dimensional representation derived from the estimated subspace. We show that our algorithm achieves the cumulative regret of $\widetilde{\mathcal{O}} \left( \max \left\{ \frac{1}{\kappa}, \sqrt{|V|} \right\} \sqrt{|V| \, T} \right)$ over time horizon $T$, where $V$ is the set of agents and $\kappa$ is a parameter dependent on the diversity of interventions. Empirical results validate that our algorithm significantly outperforms a linear bandit baseline in terms of both cumulative regret and running time.

## 1 INTRODUCTION

Social media platforms such as $\mathbb{X}$ and Facebook have become critical public infrastructures, facilitating the swift formation of public opinion. While such dynamics can serve as a powerful force for positive social change—for instance, by mobilizing collective action against undesirable political decisions—they can also exacerbate polarization and societal division (Barberá, 2020). This has driven research into algorithmic interventions to mitigate these harmful effects (Bindel et al., 2015; Matakos et al., 2017; Tu et al., 2020; Xu & Zhang, 2023; Ristache et al., 2024; Kühne et al., 2025; Liu et al., 2025; Ojer et al., 2025). A foundational model underlying this line of research is the Friedkin–Johnsen (FJ) opinion dynamics model in a social network, where each agent has two types of opinions: innate opinions, which are fixed, and expressed opinions, which evolve over time (Friedkin & Johnsen, 1990). Specifically, agents' expressed opinions evolve by taking a weighted average of their own innate opinions and their neighbors' expressed opinions. A key property that makes the FJ model particularly appealing is that the final equilibrium state of expressed opinions can be precisely calculated from the network's structure and the vector of all agents' innate opinions.

The seminal work of Musco et al. (2018) initiated the study of minimizing polarization and disagreement at the FJ model equilibrium. Polarization is defined as the extent to which expressed opinions deviate from the overall average opinion, while disagreement measures the extent to which neighboring agents hold divergent opinions. Musco et al. (2018) proposed an intervention on innate opinions, seeking to minimize the sum of polarization and disagreement by making limited adjustments to agents' innate opinions. Following this work, many researchers have explored alternative forms of intervention, including modifications to the network structure (Zhu et al., 2021) and adjustments to the strength of interpersonal connections (Cinus et al., 2023).

Despite these advances, a critical assumption persists: the full knowledge of all agents' innate opinions is available. In reality, however, acquiring this information is costly and difficult, possibly requiring extensive surveys or behavioral analysis, as highlighted by recent works (Chen et al., 2018; Chaitanya et al., 2024; Neumann et al., 2024; Cinus et al., 2025).

The foundational work of Chen et al. (2018) initiated the study of polarization and disagreement minimization in a more realistic setting by relaxing the assumption of full knowledge of agents' innate opinions; however, those innate opinions are restricted to binary values. Chaitanya et al. (2024) proposed a method to minimize an upper bound on the sum of polarization and disagreement that is valid for any possible agents' innate opinions. However, the tightness of such an upper bound to the optimum remains unclear. Moreover, the proposed method relies on semidefinite programming, which can be computationally expensive. Along similar lines, Cinus et al. (2025) considered a setting in which the innate opinions of a limited subset of nodes can be queried in order to infer the remaining ones. However, this approach still relies on the ability to query some innate opinions: in many real-world scenarios, particularly those requiring strong privacy considerations, directly querying a node's innate opinion may be very hard or simply impossible. Neumann et al. (2024) studied the estimation of relevant measures under the FJ model, such as node opinions, polarization and disagreement, without having access to the entire network and assuming to know a small number of node opinions. In particular, they showed how to estimate expressed opinions at equilibrium having access to an oracle for innate opinions, and conversely, how to estimate innate opinions from an oracle for expressed opinions at equilibrium. Their focus is on sublinear-time computability.

While highlighting the need to drop the assumption of full knowledge of innate opinions, this body of work leaves a significant, unaddressed gap: how to effectively minimize polarization and disagreement in an online setting, where agents' innate opinions are unknown and unqueryable, and must be learned through sequential observations, after each intervention. Our research directly addresses this challenge by introducing a novel framework that bridges opinion dynamics with online learning.

**Our Contributions.** In this paper, we bridge the gap by utilizing the theory of *multi-armed bandits* (Lattimore & Szepesvári, 2020). Specifically, we address the realistic online setting where agents' innate opinions are unknown and must be learned through sequential observations. This novel setting naturally mirrors the periodic nature of interventions on social media platforms. Our core contributions are as follows:

1. We formulate the above online setting as a regret minimization problem, which we term the *Online Polarization and Disagreement Minimization* (OPD-Min) problem, establishing a key connection between algorithmic interventions on social media platforms and theory of multi-armed bandits. In our formulation, at each time step, an intervention is chosen, and the learner receives only a scalar feedback representing the overall polarization and disagreement in the network. This setup, where the underlying parameters (i.e., innate opinions) are unknown and the feedback is a low-dimensional function of the intervention, naturally links the problem to the well-established stochastic low-rank matrix bandits.

2. To solve this bandit problem, we propose a two-stage algorithm. The algorithm first performs subspace estimation to identify an underlying low-dimensional structure of the problem, and then employs a linear bandit algorithm within the compact $\Theta(|V|)$ dimensional representation, which is a significant reduction from the original $|V|^2$-dimensional space, where $V$ represents the set of agents.

3. We prove that our algorithm achieves a cumulative regret bound of the form $\widetilde{\mathcal{O}}\left(\max\left\{\frac{1}{\kappa}, \sqrt{|V|}\right\}\sqrt{|V|\,T}\right)$ over any time horizon $T$, where $\kappa$ is a parameter dependent on the diversity of feasible interventions. This is the first theoretical guarantee for sequential interventions on opinion dynamics under incomplete information.

4. We substantiate our theoretical findings with extensive experiments on both synthetic and real-world networks. Our results demonstrate that our proposed algorithm significantly outperforms a linear bandit baseline in terms of both cumulative regret and running time.

**Technical Challenges.** A direct reduction to a linear bandit formulation (e.g., Abbasi-Yadkori et al. (2011)) is unsatisfactory, because the feature dimension grows quadratically with $|V|$, which results

in the regret bound that scales with $|V|^2$. On the other hand, existing low-rank matrix bandits (Lu et al., 2021; Kang et al., 2022), while seemingly well aligned with the low-rank nature of our problem, rely on sampling actions from a continuous space, such as Gaussian random matrices. This approach is infeasible to our setting whose action space is discrete, highly structured, and induced by graph Laplacians (forest matrices). As a result, existing algorithms cannot be applied directly, and their theoretical guarantees do not extend to our problem. While optimal design approaches such as (Jang et al., 2024) offer theoretical benefits, they are not directly applicable to our setting. Our action set consists of strictly symmetric forest matrices, which need not span $\mathbb{R}^{|V|^2}$. Moreover, computing the optimal design involves the covariance matrices of vectorized arms, resulting in a time complexity of $\mathcal{O}(|V|^6)$—a cost that is prohibitive for large social networks. These fundamental limitations necessitate the development of a new algorithm tailored to the unique structure of OPD-Min. See Appendix A for more detailed discussion.

## 2    BACKGROUND

In this section, we first review the Friedkin–Johnsen (FJ) opinion dynamics model (Friedkin & Johnsen, 1990), and then provide the canonical formulation of the offline problem of minimizing polarization and disagreement, which forms the basis of our online formulation.

### 2.1    FRIEDKIN–JOHNSEN (FJ) OPINION DYNAMICS MODEL

We consider a connected, undirected, edge-weighted graph $G = (V, E, w)$, where $V$ corresponds to the set of agents, $E$ represents the interactions among agents, and $w : E \to \mathbb{R}_{>0}$ quantifies the strength of the interactions. Let $\mathbf{A} = (w_{ij}) \in \mathbb{R}_{>0}^{|V| \times |V|}$ be a weighted adjacency matrix of $G$. The innate opinions of agents are represented by $\boldsymbol{s} = (s_i) \in [-1, 1]^{|V|}$, where a higher $s_i$ value represents a more favorable opinion towards a given topic. The opinion dynamics evolve in a discrete-time fashion. Specifically, agents' expressed opinions $\boldsymbol{z}^{(t)}$ at time $t + 1$ ($t = 0, 1, \dots$) are determined from $\boldsymbol{z}^{(t)}$ as follows:

$$\boldsymbol{z}^{(t+1)} = (\mathbf{D} + \mathbf{I})^{-1}(\mathbf{A}\boldsymbol{z}^{(t)} + \boldsymbol{s}) \quad (\boldsymbol{z}^{(0)} = \boldsymbol{s}),$$

where $\mathbf{D}$ is the degree matrix of $G$, whose diagonal entries are given by the weighted degrees of the nodes, and $\mathbf{I}$ is the $|V| \times |V|$ identity matrix. Since the matrix $(\mathbf{D} + \mathbf{I})^{-1}\mathbf{A}$ has a spectral radius strictly less than 1, the process converges to a unique fixed point as $t \to \infty$ (Faires & Burden, 2015, Theorem 7.17 and Lemma 7.18). Specifically, at equilibrium, the expressed opinions satisfy

$$\boldsymbol{z}^* = (\mathbf{I} + \mathbf{L})^{-1}\boldsymbol{s}, \tag{1}$$

where $\mathbf{L} = \mathbf{D} - \mathbf{A}$ is the Laplacian of $G$. Note that for a fixed network structure, the equilibrium depends only on the innate opinions $\boldsymbol{s}$. Because the spectral radius $\rho$ of $\mathbf{M} = (\mathbf{D} + \mathbf{I})^{-1}\mathbf{A}$ is strictly less than 1, the deviation from equilibrium decays at an exponential rate $\rho(\mathbf{M})^t$, so only a small number of iterations are needed for the dynamics to become arbitrarily close to $\boldsymbol{z}^*$.

The matrix $(\mathbf{I} + \mathbf{L})^{-1}$, known as the *forest matrix* (Chebotarev & Agaev, 2002), is symmetric positive definite with its eigenvalues in $(0, 1]$. Each entry $M_{ij}$ can be interpreted as the probability that a random walk starting at node $i$ is absorbed at node $j$.

### 2.2    OFFLINE PROBLEM: MINIMIZATION OF POLARIZATION AND DISAGREEMENT

The FJ model admits a rich family of quadratic functionals that capture different aspects of the equilibrium opinion landscape. These quantities are not independent but are coupled by a fundamental *conservation law* (Chen et al., 2018): minimizing one measure (e.g., disagreement) may inherently increase another (e.g., polarization), and vice versa.

In this work, we focus on two central metrics.

**Definition 1** (Polarization at equilibrium)**.** *Given an equilibrium vector $\boldsymbol{z}^* \in \mathbb{R}^{|V|}$, the* polarization *is defined as the variance of opinions around their mean:*

$$P(\boldsymbol{z}^*, G) = \sum_{i \in V}(z_i^* - \mu_{\boldsymbol{z}^*})^2, \tag{2}$$

*where $\mu_{\boldsymbol{z}^*} = \frac{1}{|V|}\sum_{i \in V} z_i^*$.*

**Assumption 1** (Innate opinions are mean-centered). *We assume that the innate opinions are mean-centered:*

$$\frac{1}{|V|} \sum_{i \in V} s_i = 0.$$

This assumption does not restrict generality; it simply removes a global offset and sets the reference point of opinions to zero. As shown in prior work (Musco et al., 2018), mean-centering is standard in the literature and does not affect polarization, disagreement, or the FJ equilibrium. It also implies that the equilibrium mean is zero, allowing polarization to simplify to $P(\boldsymbol{z}^*, G) = \|\boldsymbol{z}^*\|^2$.

**Definition 2** (Disagreement at equilibrium). *Given an equilibrium vector $\boldsymbol{z}^*$, the* disagreement *(also called external conflict) is:*

$$D(\boldsymbol{z}^*, G) = \sum_{\{i,j\} \in E} w_{ij}(z_i^* - z_j^*)^2 = (\boldsymbol{z}^*)^\top \mathbf{L} \boldsymbol{z}^*. \tag{3}$$

Polarization captures global opinion variance, while disagreement quantifies local tensions among socially connected agents. Together, they provide a comprehensive view of opinion fragmentation and are widely adopted in the literature (Musco et al., 2018; Wang & Kleinberg, 2023; Zhu et al., 2021).

Recalling that the equilibrium opinion vector $\boldsymbol{z}^*$ depends only on the innate opinions $\boldsymbol{s}$ and the Laplacian $\mathbf{L}$ via Eq. (1), one obtains the following:

**Observation 1** (Polarization plus disagreement for undirected graphs). *For any mean-centered innate opinion vector $\boldsymbol{s}$, the sum of polarization and disagreement can be written as*

$$f(\boldsymbol{s}, \mathbf{L}) = \boldsymbol{s}^\top (\mathbf{I} + \mathbf{L})^{-1} \boldsymbol{s}. \tag{4}$$

*This follows from summing Eq. (2) and Eq. (3), i.e., $f(\boldsymbol{s}, \mathbf{L}) = \boldsymbol{s}^\top (\mathbf{I} + \mathbf{L})^{-2} \boldsymbol{s} + \boldsymbol{s}^\top (\mathbf{I} + \mathbf{L})^{-1} \mathbf{L} (\mathbf{I} + \mathbf{L})^{-1} \boldsymbol{s}$, which corresponds exactly to the decomposition into polarization and disagreement.*

The canonical offline problem is to minimize Eq. (4) with respect to $\mathbf{L}$ over the set of admissible Laplacians. Prior work observed that the function $f(\mathbf{L}) = \boldsymbol{s}^\top (\mathbf{I} + \mathbf{L})^{-1} \boldsymbol{s}$ is matrix-convex whenever $\mathbf{L}$ belongs to a convex subset of Laplacians (Nordström, 2011).

Rather than assuming prior knowledge of the innate opinions $\boldsymbol{s}$ required for the one-shot offline optimization above, we adopt an *online learning* framework in which the optimal intervention is discovered through sequential interaction. We focus on online regret minimization rather than Best Arm Identification (BAI) or batched updates because interventions on platforms occur sequentially, and each action has immediate impact. Minimizing cumulative regret therefore captures the cost of suboptimal interventions, aligning with how recommender systems operate in practice. BAI or batched formulations are potential alternatives, but they assume exploration has no interim effect.

## 3 PROBLEM FORMULATION

We now formalize the problem of minimizing polarization and disagreement in an online framework, where a learner sequentially intervenes on the network and, without access to innate opinions $\boldsymbol{s}$, observes only noisy losses, i.e., noisy evaluation of polarization and disagreement. We cast the task as a stochastic low-rank matrix bandit problem.

**Online Learning Protocol.** Let $\mathcal{L} = \{\mathbf{L}_1, \mathbf{L}_2, \ldots, \mathbf{L}_K\}$ be the *intervention space*, a finite set of admissible graph Laplacians representing possible network structures. Each intervention $\mathbf{L} \in \mathcal{L}$ uniquely determines an equilibrium via its forest matrix $\mathbf{X} = (\mathbf{I} + \mathbf{L})^{-1}$. We therefore define the *action space* for our bandit algorithm as

$$\mathcal{X} = \{\mathbf{X}_i \mid \mathbf{X}_i = (\mathbf{I} + \mathbf{L}_i)^{-1}, \mathbf{L}_i \in \mathcal{L}\}.$$

By expressing the objective in Eq. (4) with the forest matrix $\mathbf{X} = (\mathbf{I} + \mathbf{L})^{-1}$ and $\boldsymbol{\Theta}^* = \boldsymbol{s}\boldsymbol{s}^\top$, we formulate it as $f(\mathbf{X}) = \langle \boldsymbol{\Theta}^*, \mathbf{X} \rangle$, where $\langle \cdot, \cdot \rangle$ denotes the trace inner product. Since $\boldsymbol{\Theta}^*$ is rank-one, the problem reduces to a low-rank matrix bandit. The online learning protocol proceeds in rounds $t = 1, \ldots, T$ as follows:

- The learner selects an intervention $\mathbf{L}_t \in \mathcal{L}$, equivalently an action $\mathbf{X}_t \in \mathcal{X}$. Under the FJ dynamics, the system converges to equilibrium, and the learner incurs the loss $f(\mathbf{X}_t)$.

- The learner observes bandit feedback in the form of a noisy loss signal: $Y_t = f(\mathbf{X}_t) + \eta_t \in \mathbb{R}$, where $\eta_t \sim N(0, \sigma^2)$. No information is revealed about the losses of other actions. For simplicity, we focus on the case where $\sigma = 1$ in the theoretical analysis.

**Objective and Regret.** The learner's goal is to minimize the *cumulative regret*, defined as the difference between the cumulative loss of the chosen actions and that of the best fixed action $\mathbf{X}^* = \arg\min_{\mathbf{X} \in \mathcal{X}} f(\mathbf{X})$:

$$R_T = \sum_{t=1}^{T} \left( f(\mathbf{X}_t) - f(\mathbf{X}^*) \right).$$

An effective algorithm must ensure that the average regret vanishes as $T \to \infty$. We refer to this complete setup as *Online Polarization and Disagreement Minimization* (OPD-Min).

**Norm Boundedness.** While norm boundedness on the unknown parameter and arms is a standard assumption in the bandit framework, a key advantage of OPD-Min is that these properties emerge naturally from its inherent structure of the FJ model.

**Unknown parameter.** Following Assumption 1, the innate opinions $s$ are mean-centered with entries bounded in $[-1, 1]$. Since $\mathbf{\Theta}^* = ss^\top$, its Frobenius norm satisfies $\|\mathbf{\Theta}^*\|_F = \|s\|_2^2 \in [0, |V|]$. To ensure the learning problem is non-trivial, we assume that the innate opinions are not all zero. For instance, assuming a constant fraction of agents have non-zero opinions gives a lower bound of $\|\mathbf{\Theta}^*\|_F = \|s\|_2^2 = \Omega(|V|)$. This prevents the degenerate case where the objective is always zero regardless of the intervention.

**Arms.** Each admissible arm corresponds to a matrix $\mathbf{X} = (\mathbf{I} + \mathbf{L})^{-1} \in \mathbb{R}^{|V| \times |V|}$, which is symmetric positive semidefinite with eigenvalues in $(0, 1]$. Consequently, $\|\mathbf{X}\|_F^2 = \sum_{i=1}^{|V|} \lambda_i(\mathbf{X})^2 \leq |V|$, where we denote the eigenvalues of a matrix $\mathbf{X}$ by $\lambda_i(\mathbf{X})$, indexed in descending order such that $\lambda_1(\mathbf{X}) \geq \lambda_2(\mathbf{X}) \geq \cdots \geq \lambda_{|V|}(\mathbf{X})$.

## 4 ALGORITHM AND REGRET ANALYSIS

Our main algorithm follows an *explore-subspace-then-refine* paradigm, adapted to the unique structure of the OPD-Min problem. While similar in structure to prior work in low-rank matrix bandits (Lu et al., 2021; Kang et al., 2022), we tailor it specifically to the rank-one case to obtain computational simplifications. We also introduce a novel analysis to handle the specific constraints imposed by our action set. The algorithm proceeds in two stages: Explore Opinion Subspace and Subspace Linear Bandit in Reduced Dimensions. Our main algorithm, *OPD-Min-ESTR*, is summarized in Algorithm 1.

### 4.1 STAGE 1: EXPLORE OPINION SUBSPACE

The initial $T_1$ rounds are dedicated to an exploration phase designed to learn the low-dimensional subspace containing the true parameter matrix $\mathbf{\Theta}^* = ss^\top$. To achieve this, we employ an estimator based on nuclear-norm regularized least-squares, a technique whose theoretical properties are thoroughly analyzed in Wainwright (2019):

$$\widehat{\mathbf{\Theta}} \in \arg\min_{\mathbf{\Theta} \in \mathbb{R}^{|V| \times |V|}} \left\{ \frac{1}{2T_1} \sum_{t=1}^{T_1} (Y_t - \langle \mathbf{X}_t, \mathbf{\Theta} \rangle)^2 + \lambda_{T_1} \|\mathbf{\Theta}\|_{\mathrm{nuc}} \right\}, \tag{5}$$

where $\lambda_{T_1}$ is the regularization parameter, which will be specified later. Here $\|\mathbf{\Theta}\|_{\mathrm{nuc}} = \sum_i \sigma_i(\mathbf{\Theta})$ is the nuclear norm, where $\sigma_i(\mathbf{\Theta})$ is the $i$-th singular value.

A key challenge, however, distinguishes our setting from conventional low-rank bandit problems (Lu et al., 2021; Kang et al., 2022). The action set $\mathcal{X}$ is composed of *forest matrices*, which are highly structured and do not allow the random sampling strategies (e.g., from a Gaussian distribution) commonly used in existing analyses that guarantee sufficient exploration. To overcome this limitation, we provide a novel theoretical analysis that explicitly leverages the *Restricted Strong Convexity (RSC)* condition for our specific set of structured actions. This analysis allows us to establish a

---

**Algorithm 1:** Explore-Subspace-Then-Refine for OPD-Min (*OPD-Min-ESTR*)

---

**Input:** Fixed arm set $\mathcal{X} = \{\mathbf{X}_i \mid \mathbf{X}_i = (\mathbf{I} + \mathbf{L}_i)^{-1}, \mathbf{L}_i \in \mathcal{L}\}$, total rounds $T$, exploration length $T_1$, regularization parameter $\lambda_{T_1}$

**Stage 1: Explore Opinion Subspace**
**for** $t = 1$ **to** $T_1$ **do**
  Pull arm $\mathbf{X}_t \in \mathcal{X}$ (e.g., uniformly at random);
  Observe the noisy loss $Y_t = f(\mathbf{X}_t) + \eta_t$
**end**
Solve the nuclear-norm penalized least-squares problem:

$$\widehat{\boldsymbol{\Theta}} = \arg\min_{\boldsymbol{\Theta} \in \mathbb{R}^{|V| \times |V|}} \frac{1}{2T_1} \sum_{t=1}^{T_1} (Y_t - \langle X_t, \boldsymbol{\Theta} \rangle)^2 + \lambda_{T_1} \|\boldsymbol{\Theta}\|_{\mathrm{nuc}}$$

Compute the top eigenvector $\hat{s}$ of $\widehat{\boldsymbol{\Theta}}$;
Extend $\hat{s}$ to its orthonormal basis: $\hat{\mathbf{S}} = [\hat{s}, \hat{\mathbf{S}}_\perp] \in \mathbb{R}^{|V| \times |V|}$;

**Stage 2: Dimensionality Reduction and Subspace Linear Bandit**
**for** $\mathbf{X} \in \mathcal{X}$ **do**
  Rotate each arm: $\mathbf{X}' := \begin{bmatrix} \hat{s} & \hat{\mathbf{S}}_\perp \end{bmatrix}^\top \mathbf{X} \begin{bmatrix} \hat{s} & \hat{\mathbf{S}}_\perp \end{bmatrix}$;
  Form reduced vectorized arm:
  $$\boldsymbol{x}' := \mathrm{vec}\left(\mathbf{X}'_{1,1}\right) \cup \mathrm{vec}(\mathbf{X}'_{2:|V|,1}) \cup \mathrm{vec}(\mathbf{X}'_{1,2:|V|}) \in \mathbb{R}^{2|V|-1}$$
**end**
Define reduced arm set $\mathcal{X}_{\mathrm{sub}} := \{\boldsymbol{x}'_1, \dots, \boldsymbol{x}'_K\} \subset \mathbb{R}^{2|V|-1}$ ;
**for** $t = T_1 + 1$ **to** $T$ **do**
  Select $\boldsymbol{x}'_t \in \mathcal{X}_{\mathrm{sub}}$ using any linear bandit algorithm with dimension $2|V| - 1$;
  Play the original arm $\mathbf{X}_t \in \mathcal{X}$ that corresponds to $\boldsymbol{x}'_t$;
  Observe the noisy loss $Y_t$;
  Update bandit algorithm with $(\boldsymbol{x}'_t, Y_t)$
**end**

---

high-probability bound on the estimation error $\|\widehat{\boldsymbol{\Theta}} - \boldsymbol{\Theta}^*\|_F$, where the number of parameters $|V|^2$ can exceed the number of samples $T_1$.

As detailed by Wainwright (2019) (e.g., Chapter 9.3.1), the RSC condition requires the loss function to have sufficient curvature only over a restricted subset of directions relevant to the true, structured parameter. For our problem, where the loss function is quadratic, the RSC condition is defined directly on the design operator. Let $\{\mathbf{X}_t\}_{t=1}^{T_1} \subset \mathcal{X}$ be the sequence of forest matrices selected during the $T_1$ exploration rounds. We define the linear observation operator $\Phi_{T_1} : \mathbb{R}^{|V| \times |V|} \to \mathbb{R}^{T_1}$ by its action on any matrix $\boldsymbol{\Delta} \in \mathbb{R}^{|V| \times |V|}$. For each $t \in T_1$, the $t$-th entry of the $T_1$-dimensional vector $\Phi_{T_1}(\boldsymbol{\Delta})$ is

$$[\Phi_{T_1}(\boldsymbol{\Delta})]_t := \langle \mathbf{X}_t, \boldsymbol{\Delta} \rangle,$$

where $[\,\cdot\,]_t$ denotes the $t$-th coordinate. With this operator, the RSC condition is stated as follows.

**Assumption 2** (RSC for Forest Matrix Sampling). *The operator $\Phi_{T_1}$ satisfies the RSC condition if there exist a curvature constant $\kappa > 0$ and a tolerance parameter $\tau_{T_1}^2 \geq 0$ such that the inequality*

$$\frac{1}{2T_1} \|\Phi_{T_1}(\boldsymbol{\Delta})\|_2^2 \geq \frac{\kappa}{2} \|\boldsymbol{\Delta}\|_F^2 - \tau_{T_1}^2 \|\boldsymbol{\Delta}\|_{\mathrm{nuc}}^2 \tag{6}$$

*holds for all matrices $\boldsymbol{\Delta}$ in the set $\mathcal{C} \subseteq \mathbb{R}^{|V| \times |V|}$. Here, $\mathcal{C}$ is the structured error set induced by nuclear-norm decomposability; see Definition 4 in Appendix B (cf. Wainwright 2019, Prop. 9.13).*

**Remark 1.** *In our setting with i.i.d. uniform draws from the fixed set $\mathcal{X}$, let $\kappa_{\min}(\mathcal{X}) := \inf_{\boldsymbol{\Delta} \in \mathcal{C} \setminus \{0\}} \frac{\frac{1}{K} \sum_{i=1}^K \langle \mathbf{X}_i, \boldsymbol{\Delta} \rangle^2}{\|\boldsymbol{\Delta}\|_F^2}$ with $K = |\mathcal{X}|$. For any $\delta \in (0, 1)$, there exists a universal constant $C >$*

0 *such that with probability at least* $1 - \delta$, *Assumption 2 holds with:* $\kappa := \kappa_{\min}(\mathcal{X})$ *and* $\tau_{T_1}^2 :=$ $\frac{C}{2}\left(\sqrt{\frac{\log(2|V|)}{T_1}} + \frac{\log(1/\delta)}{T_1}\right)$. *See Proposition 5 in Appendix B for a precise statement and proof.*

For $\boldsymbol{\Delta} := \widehat{\boldsymbol{\Theta}} - \boldsymbol{\Theta}^*$, the term $\kappa\|\boldsymbol{\Delta}\|_F^2$ guarantees that the loss function has a strong quadratic-like curvature, which is essential for ensuring that the estimator $\widehat{\boldsymbol{\Theta}}$ is close to the true parameter $\boldsymbol{\Theta}^*$. The tolerance term $\tau_{T_1}^2\|\boldsymbol{\Delta}\|_{\mathrm{nuc}}^2$ allows for this strong curvature to be violated in certain directions, but only in those directions corresponding to high-rank matrices. Since our nuclear-norm penalty specifically discourages such directions, this trade-off is manageable.

We present our first theoretical result: a high-probability bound on the Frobenius norm of the estimation error, which can be obtained via Proposition 10.6 in Wainwright (2019). The proof is provided in Appendix C. This proposition provides a guarantee that the estimation error $\|\widehat{\boldsymbol{\Theta}} - \boldsymbol{\Theta}^*\|_F^2$ decreases at a rate of $1/T_1$. This accurate estimation is the foundation upon which the efficiency of the second stage is built.

**Proposition 1** (Estimation Error Bound). *Let* $\boldsymbol{\Theta}^* = \boldsymbol{s}\boldsymbol{s}^\top \in \mathbb{R}^{|V|\times|V|}$ *be the true rank-one parameter matrix. Fix a confidence parameter* $\delta \in (0,1)$. *Define* $\widehat{\boldsymbol{\Theta}}$ *as any solution to the nuclear-norm regularized least squares problem Eq.* (5) *with* $\lambda_{T_1} = 2\sqrt{\frac{2\log(2|V|/\delta)}{T_1}}$. *Under Assumption 2, with probability at least* $1 - \delta$,

$$\|\widehat{\boldsymbol{\Theta}} - \boldsymbol{\Theta}^*\|_F^2 \leq \frac{36\,\log(2|V|/\delta)}{\kappa^2\,T_1},$$

*valid for* $128\tau_{T_1}^2 \leq \kappa$.

### 4.2 STAGE 2: DIMENSIONALITY REDUCTION AND SUBSPACE LINEAR BANDIT

After the subspace estimation phase, we reduce the original matrix bandit problem into a lower-dimensional linear bandit problem using the nuclear-norm penalized estimator $\widehat{\boldsymbol{\Theta}}$. This reduction leverages the assumed rank-one structure of the unknown parameter matrix $\boldsymbol{\Theta}^* = \boldsymbol{s}\boldsymbol{s}^\top$, which implies that the signal lies in the span of a single vector $\boldsymbol{s} \in \mathbb{R}^{|V|}$. We extract only the top singular component of $\widehat{\boldsymbol{\Theta}}$, denoted as $\widehat{\boldsymbol{s}} \in \mathbb{R}^{|V|}$, which approximates the underlying signal direction. We extend $\widehat{\boldsymbol{s}}$ to an orthonormal basis $[\widehat{\boldsymbol{s}}, \widehat{\mathbf{S}}_\perp] \in \mathbb{R}^{|V|\times|V|}$ to define a rotation matrix.

For each matrix arm $\mathbf{X} \in \mathcal{X} \subset \mathbb{R}^{|V|\times|V|}$, we then perform the bilinear rotation:

$$\mathbf{X}' := \begin{bmatrix} \widehat{\boldsymbol{s}} & \widehat{\mathbf{S}}_\perp \end{bmatrix}^\top \mathbf{X} \begin{bmatrix} \widehat{\boldsymbol{s}} & \widehat{\mathbf{S}}_\perp \end{bmatrix}.$$

Then, we discard the bottom-right block $\mathbf{X}'_{2:|V|,2:|V|}$ corresponding to directions orthogonal to the estimated subspace. We then form a reduced arm vector in $\mathbb{R}^{2|V|-1}$ by concatenating the top-left scalar, the first column below the diagonal, and the first row to the right of the diagonal:

$$\boldsymbol{x}_{\mathrm{sub}}(\mathbf{X}) := \begin{bmatrix} \mathbf{X}'_{1,1} \\ \mathbf{X}'_{2:|V|,1} \\ \mathbf{X}'_{1,2:|V|} \end{bmatrix} \in \mathbb{R}^{2|V|-1}.$$

We call $k := 2|V| - 1$ as the projected dimension. This transformation defines a fixed, low-dimensional arm set over which we can run a standard linear bandit algorithm (e.g., (Abbasi-Yadkori et al., 2011; Dani et al., 2008) or see also (Lattimore & Szepesvári, 2020)) for the remaining $T_2 = T - T_1$ rounds. The computational cost of computing $\widehat{\boldsymbol{s}}$ is low, as we only need the top eigenvector of a symmetric matrix, which can be obtained via the power method or Lanczos iteration in time $\mathcal{O}(|V|^2/\varepsilon)$ for accuracy $\varepsilon > 0$. This projection reduces the ambient dimension from $|V|^2$ to $\mathcal{O}(|V|)$, enabling faster convergence and sharper confidence bounds.

### 4.3 REGRET ANALYSIS

The overall regret decomposes into a projection error due to misalignment of $\widehat{\boldsymbol{s}}$ and $\boldsymbol{s}$, which vanishes with large $T_1$, and the standard regret from the linear bandit phase scaling with $\sqrt{T_2}$. The following theorem formalizes the resulting regret bound. The proof is provided in Appendix D.

**Theorem 4.1** (Regret Bound for *OPD-Min-ESTR*). *Suppose the subspace linear bandit algorithm used in Stage 2 enjoys $R_T^{\text{sub}} = \tilde{\mathcal{O}}(k\sqrt{T})$, where $k = 2|V| - 1$. Under Assumption 2, for any failure probability $\delta \in (0,1)$ and an optimally chosen number of exploration rounds $T_1 \asymp \frac{1}{\|\boldsymbol{s}\|^2 \kappa} \sqrt{T \log(2|V|/\delta)}$, the total regret of OPD-Min-ESTR over sufficiently large $T \geq T_0 \asymp \frac{\|\boldsymbol{s}\|^4}{\kappa^2} \log(2|V|/\delta)$ rounds satisfies, with probability at least $1 - \delta$,*

$$R_T = \tilde{\mathcal{O}}\left(\max\left\{\frac{1}{\kappa}, \sqrt{|V|}\right\}\sqrt{|V| \cdot T}\right),$$

The regret bound in Theorem 4.1 confirms the statistical efficiency of *OPD-Min-ESTR*. The $\tilde{\mathcal{O}}(\sqrt{T})$ rate of the time horizon $T$ is optimal for stochastic bandit problems (Lattimore & Szepesvári, 2020), and the dependence on $|V|$ instead of $|V|^2$ demonstrates the effectiveness of the two-stage approach.

The following corollary specifies the regret bound when the algorithm's hyperparameter is set using a more practically available lower bound, $\ell_s$, instead of the unknown true signal strength $\|\boldsymbol{s}\|^2$.

**Corollary 1** (Regret Bound with a Lower Bound on Signal Strength). *Under the same assumptions as Theorem 4.1, suppose that the true signal strength $\|\boldsymbol{s}\|^2$ is unknown, but a lower bound $\ell_s > 0$ is known, such that $\|\boldsymbol{s}\|^2 \geq \ell_s$. If we set the exploration phase length $T_1 \asymp \frac{1}{\ell_s \kappa} \sqrt{T \log \frac{2|V|}{\delta}}$, then the total regret $R_T$ is bounded by:*

$$R_T = \tilde{\mathcal{O}}\left(\frac{1}{\kappa}\sqrt{|V| \cdot T}\right).$$

**Remark 2.** *The curvature parameter $\kappa$ in our RSC condition is set to $\kappa_{\min}(\mathcal{X})$. This quantity serves as a crucial measure of the diversity of the available actions strictly along the error manifold defined by the cone $\mathcal{C}$. In our framework, this parameter is critical; as our error bounds scale with $1/\kappa^2$ and the regret bound scales with $1/\kappa$, a small $\kappa$ implies a weak theoretical guarantee. However, for highly structured problems such as rank-one matrix recovery, this global metric can be overly pessimistic. We complement the theory with an empirical study in Appendix G.1, which shows that the effective curvature in practice can be substantially higher than the worst-case over the cone suggested by $\kappa_{\min}(\mathcal{X})$.*

## 5 EXPERIMENTS

In this section, we evaluate the performance of our algorithm, which integrates OFUL (Abbasi-Yadkori et al., 2011) as the Stage-2 optimizer in a $(2|V| - 1)$-dimensional subspace. The benchmarks consist of: (i) a high-dimensional OFUL baseline that operates directly in the $|V|^2$-dimensional space, and (ii) an oracle subspace variant that has access to the true subspace $\boldsymbol{\Theta}^*$ and thus provides a natural lower bound on the regret achievable during the exploration phase.

All experiments follow the online protocol described in Section 3. At each round $t$, the learner selects an arm $\mathbf{X}_t$ from a finite set and observes the noisy scalar loss $Y_t = \langle \mathbf{X}_t, \boldsymbol{\Theta}^* \rangle + \eta_t$, where $\eta_t \sim \mathcal{N}(0, \sigma_\eta^2)$. The objective is to minimize cumulative polarization-plus-disagreement, measured in terms of regret relative to the best fixed arm in hindsight. We first report results in a controlled low-rank bandit environment, consistent with prior works (Lu et al., 2021; Kang et al., 2022). Additional experiments on real-world networks, scalability, and sensitivity are provided in Appendix G.

**Experimental Setup.** We consider $|\mathcal{X}| \in \{10, 100, 1000\}$ candidate arms. Each arm is constructed by perturbing a fixed undirected Laplacian $\mathbf{L}$ with $|V|$ random rank-one updates, generalizing the construction of (Zhu et al., 2021); and each perturbation weight is sampled uniformly from $[0.5, 1.5]$. This produces a low-diversity action space, corresponding to a worst-case setting for our algorithm.

The innate opinion vector $\boldsymbol{s}$ is sampled uniformly from $[-1, 1]^{|V|}$ and then mean-centered, following standard practice (Musco et al., 2018). Environment noise is set to $\sigma_\eta \in \{0.01, 0.1, 1.0\}$. We fix the confidence parameter to $\delta = 0.001$ and the OFUL regularization to $\lambda = 0.1$. The time horizon is $T = 10{,}000$, with our algorithm allocating $T_1 = \sqrt{T}$ rounds to subspace exploration and $T_2 = T - T_1$ rounds to projected OFUL. By contrast, the high-dimensional OFUL baseline uses the entire horizon. Each experiment is repeated over 100 independent runs. We report mean cumulative regret with one standard deviation, together with average runtime.

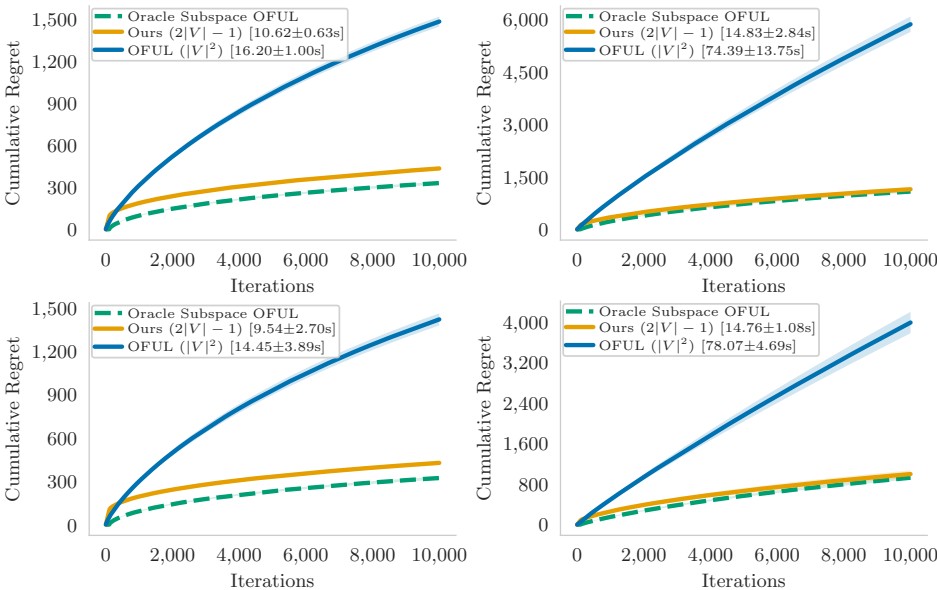

Figure 1: Cumulative regret for Erdős–Rényi graphs (top) and homophilic Stochastic Block Model graphs (bottom) with $|V| \in \{8, 16\}$. Runtime (mean ± std) over 100 repetitions is reported in the legend. For ER graphs the edge probability is $p = 0.2$. For SBM graphs, two communities are generated with sizes $|V_1| \approx 0.75|V|$, $|V_2| = |V| - |V_1|$, intra-community edge probability $p = 0.5$, and inter-community probability $p = 0.07$.

**Results.** Figure 1 compares OPD-Min, full-dimensional OFUL, and the subspace oracle. Across both network models and sizes ($|V| = 8, 16$), OPD-Min consistently achieves lower regret and faster runtime than OFUL. At $|V| = 16$, the differences become particularly pronounced: OFUL suffers both substantially higher regret and significantly slower execution. In contrast, OPD-Min closely tracks the oracle baseline, effectively closing the initial gap due to subspace estimation. These results demonstrate that exploiting the low-rank structure of $\Theta^*$ yields substantial improvements in both sample efficiency and computational efficiency.

**Additional Experiments.** Further results are provided in the appendix and demonstrate: (i) empirical lower bounds for the restricted strong convexity (RSC) condition (Sec. G); (ii) scalability to large graphs with up to $|V| = 1024$ nodes (Sec. G.2); (iii) applications to real-world graphs, including Florentine families, Davis Southern women, Karate club, and Les Misérables (Sec. G.4); and (iv) robustness through sensitivity analyses on noise levels and the number of arms (Sec. G.5).

## 6 CONCLUSION

We introduced the first formalization of minimizing polarization and disagreement in the Friedkin–Johnsen model under incomplete information in an online setting. This setting naturally mirrors the continuous interventions observed on real platforms and the dynamic nature of opinion formation.

We cast the problem as regret minimization in stochastic low-rank matrix bandits, where after each intervention the opinion dynamics are allowed to converge to equilibrium, and the learner observes only a scalar feedback corresponding to the resulting polarization and disagreement. Throughout the process, the innate opinions of agents remain unknown. To address this, we proposed the novel two-stage algorithm *OPD-Min-ESTR*: first estimating the latent subspace, then running a linear bandit method in a compact $2|V| - 1$ dimensional representation derived from the estimate.

By leveraging structural properties of opinion dynamics and tools from matrix analysis and bandit optimization, we proved that the algorithm achieves $\widetilde{O}(|V|\sqrt{T})$ regret under mild diversity assumptions of feasible interventions. Experiments on synthetic and real graphs confirmed both the statistical and computational benefits of our approach over full-dimensional baselines.

This work opens several promising directions for future work. On the theoretical side, developing problem-specific notions of curvature could sharpen our guarantees. Our analysis relies on a global RSC condition which, though sufficient, may be overly conservative in the rank-one setting. Exploring effective curvature localized to the relevant error manifold could yield tighter guarantees even when the global curvature $\kappa_{\min}(\mathcal{X})$ is small. On the empirical side, a natural next step is to apply the framework to real-world opinion dynamics, leveraging observational or intervention data from online platforms to capture the complexities of agent behavior and feedback. Moreover, moving beyond scalar equilibrium feedback to richer but noisy signals (e.g., community-level polarization) could further bridge the gap between theoretical analysis and practical deployment.

## ACKNOWLEDGMENTS

This work is partially supported by Japan Science and Technology Agency (JST) Strategic Basic Research Programs PRESTO "R&D Process Innovation by AI and Robotics: Technical Foundations and Practical Applications" grant number JPMJPR24T2.

## REPRODUCIBILITY STATEMENT

All theorems and lemmas are stated under consistent notation in Sec. 4, and complete proofs are provided in the appendix. In particular, Sec. B establishes the restricted strong convexity (RSC) conditions, Sec. C derives the estimation error bound for the unknown parameter in Stage 1, and Sec. D presents the main regret analysis. The proposed algorithm is described in Alg. 1, and the implementation is available at `https://github.com/FedericoCinus/online-min-pol`. A code overview is provided in Sec. E.1. For empirical evaluation, Sec. E.2 details the experimental setup, including parameters and datasets (all publicly available), and Sec. F provides additional details on the benchmark algorithms.

## ETHICS STATEMENT

**Societal Impact.** This work develops a framework for designing interventions aimed at reducing opinion polarization in online social networks. The intended beneficiaries are platform users and society at large, as mitigating extreme fragmentation can support more constructive dialogue, improve information flow, and reduce the likelihood of harmful echo chambers. By grounding the analysis in a well-established opinion-dynamics model, the approach provides a transparent mechanism for understanding how small structural changes in a network can influence collective outcomes.

**Potential Misuse and Risks.** Despite its positive intent, any methodology that models or modifies opinions carries inherent risks. One potential misuse is the deployment of such tools to intentionally increase polarization, manipulate user beliefs, or disproportionately influence specific demographic groups. Mathematically, the same optimization framework could be applied to maximize polarization simply by changing the sign of the objective. However, doing so would require privileged access to platform-level controls (e.g., recommendation, exposure, or graph interventions), which are typically restricted to platform operators rather than arbitrary malicious actors.

Our formulation assumes that agents' innate opinions are unknown and never directly observed; the learner only receives a scalar feedback, corresponding to the global polarization-plus-disagreement value at equilibrium. Thus, the framework does not rely on individual-level opinion estimation, but instead on aggregated network-level outcomes. Nevertheless, repeated interventions that affect exposure patterns can still have normative implications (e.g., shaping which content or connections are promoted), and such decisions should be subject to appropriate ethical and governance safeguards.

**Beneficiaries vs. Risks.** The primary beneficiaries of responsible use are online communities where reduced polarization leads to healthier discourse, more diverse exposure, and improved collective decision-making. The main risks arise if interventions are applied without transparency or accountability, potentially impacting user autonomy or systematically affecting certain groups more than others. These risks highlight the need for careful oversight: any real-world deployment should comply with regulatory standards, respect user rights, and be designed with fairness and explainability in mind.

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

# Appendix

## A    COMPARISON WITH EXISTING LINEAR OR LOW RANK MATRIX BANDITS

To address our formulation, one might consider adapting standard bandit frameworks (Lattimore & Szepesvári, 2020). A naive approach is to linearize the objective by treating it as an inner product in an $|V|^2$-dimensional space, i.e., $\text{vec}(\boldsymbol{s}\boldsymbol{s}^\top)^\top \text{vec}(\mathbf{X})$, where $\boldsymbol{s}$ is the innate opinions vector and $\mathbf{X}$ is a matrix determined by the intervention. This reduces the problem to a linear bandit (e.g., (Abbasi-Yadkori et al., 2011)), but at the cost of a prohibitive regret bound of $\tilde{O}(|V|^2\sqrt{T})$, rendering it impractical even for small networks.

An alternative, seemingly more suitable approach would be to leverage low-rank matrix bandits, as the unknown parameter matrix $\boldsymbol{s}\boldsymbol{s}^\top$ is inherently rank-one. A broad line of work studies bandit and estimation problems under low-rank structure (Katariya et al., 2017; Jun et al., 2019; Huang et al., 2021; Jang et al., 2021; Lu et al., 2021; Kang et al., 2022; Jang et al., 2024; Kang et al., 2024; Yi et al., 2024; Wang et al., 2025). All of them share the principle that a large parameter matrix can be effectively approximated by a low-rank representation, enabling sample efficiency and computational tractability.

Our two-stage, Explore-Subspace-Then-Refine (ESTR) type algorithm belongs to a successful line of research on efficient low-rank matrix bandits, drawing inspiration from seminal works such as Jun et al. (2019); Lu et al. (2021); Kang et al. (2022); Jang et al. (2024). All these works share a common high-level strategy: an initial exploration phase to estimate the latent low-rank subspace, followed by an exploitation phase that leverages this structure to solve a lower-dimensional bandit problem such as OFUL (Abbasi-Yadkori et al., 2011). The first ESTR algorithm by Jun et al. (2019) was designed for bilinear bandits where arms are rank-one matrices: the learner chooses a pair of arms, each from two different action spaces of dimension $d_1$ and $d_2$.

Our algorithm is built upon efficient low-rank matrix bandits, most notably the LowESTR algorithm of Lu et al. (2021). LowESTR first performs an exploration phase using nuclear norm regularization to recover a low-rank structure, and then runs a linear bandit algorithm restricted to the estimated subspace. Kang et al. (2022) present frameworks (G-ESTT, G-ESTS) that extend low-rank matrix bandits to generalized linear models, which use Stein's method for matrix estimation during the exploration phase, followed by a refinement phase that runs a linear bandit in a reduced-dimensional space. They provide regret bounds of order $\widetilde{\mathcal{O}}(\sqrt{(d_1 + d_2)^3 rT}/D_r)$ for the general rank $r$ matrix of dimensions $d_1$ and $d_2$, where $D_r$ is the $r$-th largest singular value for the true matrix $\boldsymbol{\Theta}^*$. Jang et al. (2024) propose LowPopArt for low-rank trace regression with experimental designs, and its arm set geometry-adaptive bandit algorithms LPA-ETC and LPA-ESTR using LowPopArt.

However, these existing works cannot be applied directly to our problem, where the action set $\mathcal{X}$ consists of highly structured forest matrices derived from graph Laplacians. In Lu et al. (2021); Kang et al. (2022), they posit the existence of a nice exploration distribution over the arm set that ensures sufficient exploration (e.g., Assumption 2 in (Lu et al., 2021)). Specifically, Lu et al. (2021) assume the existence of an exploration distribution $D$ over the action set $\mathcal{X}$, whose covariance has $\lambda_{\min}(\Sigma) \gtrsim 1/(d_1 d_2)$ and is sub-Gaussian. This assumption naturally holds when $\text{conv}(\mathcal{X})$ contains a ball of radius $R = O(1)$, e.g., in continuous and isotropic settings. In contrast, our setting involves a finite and highly structured action set (forest matrices), where such a distribution is not available. Recent work, such as LowPopArt by Jang et al. (2024), develops a theoretical framework applicable to general arm sets via optimal experimental design. More recently, Lee et al. (2026) extended this framework to generalized linear models. However, these approaches require constructing and inverting a covariance matrix in the vectorized space of dimension $|V|^2 \times |V|^2$. As a consequence, forming the corresponding estimator entails a computational cost on the order of $\mathcal{O}(|V|^6)$ in the worst case. This scaling is prohibitive in social network settings with many agents. Also, in those works, the action space is supposed to span the full space, while our restricted actions derived from undirected graph Laplacians do not span in $\mathbb{R}^{|V|^2}$, which implies we cannot rely on the existence of a well-behaved exploration distribution. While one could potentially adapt these methods by restricting the design to the span of symmetric actions, developing such a variant is beyond our scope.

To overcome these limitations, our work provides a novel and self-contained analysis. Instead of assuming favorable sampling properties, we directly prove that the Restricted Strong Convexity (RSC) condition holds for our specific action set under uniform sampling (see Proposition 5 and its proof in Appendix B). This analysis validates our exploration phase without external assumptions about the

action set structure, establishing a solid foundation for using the explore-then-refine framework in online opinion dynamics.

# B    RSC Condition

## B.1    Preliminaries

In order to analyze the RSC condition for our case, we follow the fundamental tools in Section 10.2.1 of Wainwright (2019), which deals with the general rank $r$ case. For the special case where the true matrix is $\mathbf{\Theta}^* = \boldsymbol{s}\boldsymbol{s}^\top$, the left and right singular vector subspaces are identical: $\mathcal{U}_1 = \mathcal{V}_1 = \operatorname{span}(\boldsymbol{s})$.

**Definition 3** (Subspaces for $\mathbf{\Theta}^* = \boldsymbol{s}\boldsymbol{s}^\top$). *For the rank-one matrix $\mathbf{\Theta}^* = \boldsymbol{s}\boldsymbol{s}^\top$, the key subspaces are defined as follows:*

- *The model subspace $\mathcal{M}$ consists of all matrices that are scalar multiples of $\mathbf{\Theta}^*$:*

$$\mathcal{M} := \left\{ \mathbf{\Delta} \in \mathbb{R}^{|V| \times |V|} \,\middle|\, \mathbf{\Delta} = c \cdot \boldsymbol{s}\boldsymbol{s}^\top \text{ for some scalar } c \in \mathbb{R} \right\}. \tag{7}$$

- *The perturbation subspace $\mathcal{M}^\perp$ consists of all matrices that are orthogonal to $\boldsymbol{s}$ in both their row and column spaces:*

$$\mathcal{M}^\perp := \left\{ \mathbf{\Delta} \in \mathbb{R}^{|V| \times |V|} \,\middle|\, \mathbf{\Delta}\boldsymbol{s} = \mathbf{0} \text{ and } \boldsymbol{s}^\top\mathbf{\Delta} = \mathbf{0}^\top \right\}. \tag{8}$$

- *The decomposability subspace $\overline{\mathcal{M}} = (\mathcal{M}^\perp)^\perp$ is the set of all matrices that can be written as the sum of a matrix whose column space is in $\operatorname{span}(\boldsymbol{s})$ and a matrix whose row space is in $\operatorname{span}(\boldsymbol{s})$:*

$$\overline{\mathcal{M}} := \left\{ \mathbf{\Delta} \in \mathbb{R}^{|V| \times |V|} \,\middle|\, \mathbf{\Delta} = \boldsymbol{s}\boldsymbol{a}^\top + \boldsymbol{b}\boldsymbol{s}^\top \text{ for some vectors } \boldsymbol{a}, \boldsymbol{b} \in \mathbb{R}^{|V|} \right\}. \tag{9}$$

The following proposition, adapted from Proposition 9.13 of Wainwright (2019) for specialization for nuclear norm regularization and our specific case, establishes that for a suitable choice of the regularization parameter $\lambda_{T_1}$, the estimation error vector is not arbitrary but is confined to a specific cone-like set.

**Proposition 2** (cf. Prop. 9.13 of Wainwright (2019)). *Let $L_{T_1} : \Omega \to \mathbb{R}$ be a convex function and let the regularizer be the nuclear norm. Consider the subspace pair $(\mathcal{M}, \overline{\mathcal{M}})$ over which the nuclear norm is decomposable. Then conditioned on the event*

$$G(\lambda_{T_1}) := \left\{ \|\nabla L_{T_1}(\mathbf{\Theta}^*)\|_{\operatorname{op}} \leq \frac{\lambda_{T_1}}{2} \right\}, \tag{10}$$

*any optimal solution $\widehat{\mathbf{\Theta}}$ yields an error vector $\mathbf{\Delta} = \widehat{\mathbf{\Theta}} - \mathbf{\Theta}^*$ that belongs to the set:*

$$\mathcal{C}_{\mathbf{\Theta}^*} := \left\{ \mathbf{\Delta} \in \Omega \,\middle|\, \|\mathbf{\Delta}_{\overline{\mathcal{M}}^\perp}\|_{\operatorname{nuc}} \leq 3\|\mathbf{\Delta}_{\overline{\mathcal{M}}}\|_{\operatorname{nuc}} + 4\|\mathbf{\Theta}^*_{\mathcal{M}^\perp}\|_{\operatorname{nuc}} \right\}. \tag{11}$$

*In the well-specified case where the true parameter $\mathbf{\Theta}^*$ belongs to the model subspace $\mathcal{M}$, the approximation error term $\|\mathbf{\Theta}^*_{\mathcal{M}^\perp}\|_{\operatorname{nuc}}$ vanishes. In this scenario, the set simplifies to the cone $\mathcal{C}$, and the error vector is guaranteed to satisfy $\|\mathbf{\Delta}_{\overline{\mathcal{M}}^\perp}\|_{\operatorname{nuc}} \leq 3\|\mathbf{\Delta}_{\overline{\mathcal{M}}}\|_{\operatorname{nuc}}$.*

Recalling that an optimal solution $\widehat{\mathbf{\Theta}}$ is symmetric, this provides a revised RSC condition.

**Definition 4** (RSC Condition Restricted to the Cone). *An observation operator $\Phi_{T_1}$ is said to satisfy the Restricted Strong Convexity (RSC) condition if there exist a curvature constant $\kappa > 0$ and a tolerance $\tau_{T_1}^2 \geq 0$ such that the following inequality holds for all matrices $\mathbf{\Delta}$ belonging to the cone $\mathcal{C}$:*

$$\frac{1}{2T_1} \|\Phi_{T_1}(\mathbf{\Delta})\|_2^2 \geq \frac{\kappa}{2} \|\mathbf{\Delta}\|_F^2 - \tau_{T_1}^2 \|\mathbf{\Delta}\|_{\operatorname{nuc}}^2, \tag{12}$$

*where $\mathcal{C} := \left\{ \mathbf{\Delta} \in \mathbb{S}^{|V| \times |V|} \,\middle|\, \|\mathbf{\Delta}_{\overline{\mathcal{M}}^\perp}\|_{\operatorname{nuc}} \leq 3\|\mathbf{\Delta}_{\overline{\mathcal{M}}}\|_{\operatorname{nuc}} \right\}$.*

### B.2 Auxiliary results

We need to introduce the Talagrand concentration for empirical processes.

**Proposition 3** (Theorem 3.27 of Wainwright (2019)). *Consider a countable class of functions $\mathcal{F} := \{f_\theta : \theta \in \Theta\}$ for all $\theta$ uniformly bounded by $b$, i.e., $\sup_\theta \|f_\theta\|_\infty \leq b$. Let $Z = \sup_{\theta \in \Theta} |\frac{1}{T_1} \sum_{i=1}^{T_1} f_\theta(X_i)|$. Assume that for some constants $\sigma^2$, we have $\sup_\theta \mathbb{E}[f_\theta(X)^2] \leq \sigma^2$ Then for all $t > 0$, the random variable $Z$ satisfies the upper tail bound*

$$\mathbb{P}\left(Z \geq 2\mathbb{E}[Z] + t\right) \leq \exp\left(-\frac{T_1 t^2}{8\sigma^2 + 4bt}\right).$$

**Lemma 1** (cf. Proposition. 4.11 of Wainwright (2019) ). *Let $\mathcal{F}$ be a class of functions. For an i.i.d. sequence $\{X_i\}_{i=1}^{T_1}$, the expected supremum of the centered empirical process is bounded by twice the expected supremum of the Rademacher process:*

$$\mathbb{E}\left[\sup_{f \in \mathcal{F}} \left|\frac{1}{T_1} \sum_{i=1}^{T_1} (g(X_i) - \mathbb{E}[g(X)])\right|\right] \leq 2\mathbb{E}\left[\sup_{f \in \mathcal{F}} \left|\frac{1}{T_1} \sum_{i=1}^{T_1} \varepsilon_i g(X_i)\right|\right],$$

*where $\{\varepsilon_i\}_{i=1}^{T_1}$ is an i.i.d. Rademacher sequence (taking values in $\{-1, +1\}$ with equal probability), independent of $\{X_i\}_{i=1}^{T_1}$.*

*Proof.* Let $\{Y_i\}_{i=1}^{T_1}$ be an i.i.d. sequence of random variables, drawn from the same distribution as $\{X_i\}_{i=1}^{T_1}$ and independent of it. As $\mathbb{E}[g(X_i)] = \mathbb{E}_Y[g(Y_i)]$ for any sample $i \in [T_1]$, we have

$$Z := \mathbb{E}_X\left[\sup_{f \in \mathcal{F}} \left|\frac{1}{T_1} \sum_{i=1}^{T_1} (g(X_i) - \mathbb{E}[g(X_i)])\right|\right] = \mathbb{E}_X\left[\sup_{f \in \mathcal{F}} \left|\frac{1}{T_1} \sum_{i=1}^{T_1} (g(X_i) - \mathbb{E}_Y[g(Y_i)])\right|\right].$$

As $\sup(\cdot)$ is a convex function and by Jensen's inequality,

$$Z = \mathbb{E}_X\left[\sup_{f \in \mathcal{F}} \left|\mathbb{E}_Y\left[\frac{1}{T_1} \sum_{i=1}^{T_1} (g(X_i) - g(Y_i))\right]\right|\right]$$

$$\leq \mathbb{E}_X \mathbb{E}_Y\left[\sup_{f \in \mathcal{F}} \left|\frac{1}{T_1} \sum_{i=1}^{T_1} (g(X_i) - g(Y_i))\right|\right].$$

Since i.i.d. Rademacher random variables $\{\varepsilon_i\}_{i=1}^{T_1}$ are independent of both $\{X_i\}$ and $\{Y_i\}$, we have

$$Z \leq \mathbb{E}_{X,Y,\varepsilon}\left[\sup_{f \in \mathcal{F}} \left|\frac{1}{T_1} \sum_{i=1}^{T_1} \varepsilon_i(g(X_i) - g(Y_i))\right|\right].$$

Using the triangle inequality, we further obtain:

$$Z \leq \mathbb{E}_{X,Y,\varepsilon}\left[\sup_{f \in \mathcal{F}} \left(\left|\frac{1}{T_1} \sum_{i=1}^{T_1} \varepsilon_i g(X_i)\right| + \left|\frac{1}{T_1} \sum_{i=1}^{T_1} \varepsilon_i(-g(Y_i))\right|\right)\right]$$

$$= \mathbb{E}_{X,\varepsilon}\left[\sup_{f \in \mathcal{F}} \left|\frac{1}{T_1} \sum_{i=1}^{T_1} \varepsilon_i g(X_i)\right|\right] + \mathbb{E}_{Y,\varepsilon}\left[\sup_{f \in \mathcal{F}} \left|\frac{1}{T_1} \sum_{i=1}^{T_1} \varepsilon_i g(Y_i)\right|\right]$$

$$\leq 2\mathbb{E}_{X,\varepsilon}\left[\sup_{f \in \mathcal{F}} \left|\frac{1}{T_1} \sum_{i=1}^{T_1} \varepsilon_i g(X_i)\right|\right],$$

which completes the proof.

$\square$

Next, we introduce the Matrix Rademacher Series.

**Proposition 4** (Theorem 4.1.1 of Tropp (2012)). *Consider a finite sequence $\{\mathbf{B}_k\}$ of fixed symmetric matrices with dimension $d \times d$, and let $\{\zeta_k\}$ be a finite sequence of independent Rademacher random variables. Let $v := \|\sum_{k=1}^{m} \mathbf{B}_k \mathbf{B}_k^\top\|_{op}$. Then, $\mathbb{E}\|\sum_k \zeta_k \mathbf{B}_k\| \leq \sqrt{2v \cdot \log(2d)}$.*

### B.3 PROPOSITION 5 AND ITS PROOF

**Proposition 5** (RSC Condition for Uniform Sampling over Forest Matrices). *Let the observation operator $\Phi_{T_1}$ be constructed from $T_1$ i.i.d. samples drawn uniformly from a fixed set of measurement matrices $\{\mathbf{X}_i\}_{i=1}^K \subset \mathbb{R}^{|V| \times |V|}$. Then, for any failure probability $\delta \in (0,1)$, there exists a universal constant $C > 0$ such that with probability at least $1 - \delta$, the RSC condition from Definition 4 holds for all matrices $\mathbf{\Delta}$ in the cone $\mathcal{C} \setminus \{0\}$. The curvature and tolerance parameters are given by:*

$$\kappa := \kappa_{\min}(\mathcal{X}) \quad and \quad \tau_{T_1}^2 := \frac{C}{2} \left( \sqrt{\frac{\log(2|V|)}{T_1}} + \frac{\log(1/\delta)}{T_1} \right)$$

*where $\kappa_{\min}(\mathcal{X}) := \inf_{\mathbf{\Delta} \in \mathcal{C} \setminus \{0\}} \frac{\frac{1}{K} \sum_{i=1}^K \langle \mathbf{X}_i, \mathbf{\Delta} \rangle^2}{\|\mathbf{\Delta}\|_F^2}$.*

*Proof of Proposition 5.* The proof is based on bounding the deviation between the sample covariance matrix $\hat{\mathbf{H}} := \frac{1}{T_1} \sum_{t=1}^{T_1} \boldsymbol{x}_t \boldsymbol{x}_t^\top$ and its expectation $\bar{\mathbf{H}} := \mathbb{E}[\hat{\mathbf{H}}]$, where $\boldsymbol{x}_t = \mathrm{vec}(\mathbf{X}_t)$.

To verify the RSC condition, we evaluate the quadratic form for any matrix $\mathbf{\Delta}$ within the cone $\mathcal{C}$:

$$\frac{1}{T_1} \|\Phi_{T_1}(\mathbf{\Delta})\|_2^2 = \mathrm{vec}(\mathbf{\Delta})^\top \hat{\mathbf{H}} \, \mathrm{vec}(\mathbf{\Delta}) = \underbrace{\mathrm{vec}(\mathbf{\Delta})^\top \bar{\mathbf{H}} \, \mathrm{vec}(\mathbf{\Delta})}_{\text{(I) Population Curvature}} + \underbrace{\mathrm{vec}(\mathbf{\Delta})^\top (\hat{\mathbf{H}} - \bar{\mathbf{H}}) \, \mathrm{vec}(\mathbf{\Delta})}_{\text{(II) Statistical Deviation}}. \tag{13}$$

Using the definition of $\kappa_{\min}(\mathcal{X})$, for any $\mathbf{\Delta} \in \mathcal{C}$, we immediately have:

$$\mathrm{vec}(\mathbf{\Delta})^\top \bar{\mathbf{H}} \, \mathrm{vec}(\mathbf{\Delta}) \geq \kappa_{\min}(\mathcal{X}) \|\mathbf{\Delta}\|_F^2. \tag{14}$$

Suppose that with probability at least $1 - \delta$,

$$\left| \mathrm{vec}(\mathbf{\Delta})^\top (\hat{\mathbf{H}} - \bar{\mathbf{H}}) \, \mathrm{vec}(\mathbf{\Delta}) \right| \leq C \, \|\mathbf{\Delta}\|_{\mathrm{nuc}}^2 \left( \sqrt{\frac{\log(2|V|)}{T_1}} + \frac{\log(1/\delta)}{T_1} \right), \tag{15}$$

which will be proved later in Proposition 6. We substitute these bounds into Eq. (13). Then, with probability at least $1 - \delta$, we obtain

$$\frac{1}{2T_1} \|\Phi_{T_1}(\mathbf{\Delta})\|_2^2 \geq \frac{1}{2} \left( \kappa_{\min}(\mathcal{X}) \|\mathbf{\Delta}\|_F^2 \right) - \frac{1}{2} \left( C \, \|\mathbf{\Delta}\|_{\mathrm{nuc}}^2 \left( \sqrt{\frac{\log(2|V|)}{T_1}} + \frac{\log(1/\delta)}{T_1} \right) \right)$$

$$= \frac{\kappa_{\min}(\mathcal{X})}{2} \|\mathbf{\Delta}\|_F^2 - \left( \frac{C}{2} \left( \sqrt{\frac{\log(2|V|)}{T_1}} + \frac{\log(1/\delta)}{T_1} \right) \right) \|\mathbf{\Delta}\|_{\mathrm{nuc}}^2.$$

The last step to prove Eq. (15) is due to the following proposition, which completes the proof. $\square$

**Proposition 6.** *Let the assumptions of Definition 4 hold. There exists a universal constant $C > 0$ such that for any $\delta \in (0,1)$, the following inequality holds with probability at least $1 - \delta$ for all matrices $\mathbf{\Delta} \in \mathcal{C}$ simultaneously:*

$$\left| \mathrm{vec}(\mathbf{\Delta})^\top (\hat{\mathbf{H}} - \bar{\mathbf{H}}) \, \mathrm{vec}(\mathbf{\Delta}) \right| \leq C \, \|\mathbf{\Delta}\|_{\mathrm{nuc}}^2 \left( \sqrt{\frac{\log(2|V|)}{T_1}} + \frac{\log(1/\delta)}{T_1} \right).$$

*Proof of Proposition 6.* For each $\mathbf{\Delta} \in \mathcal{C}$, define the zero-mean function $f_{\mathbf{\Delta}}(\mathbf{X}) := \langle \mathbf{X}, \mathbf{\Delta} \rangle^2 - \mathbb{E}[\langle \mathbf{X}, \mathbf{\Delta} \rangle^2]$. For term (II) in Eq. (13), we first expand the statistical deviation term for a matrix $\mathbf{\Delta}$:

$$\nu(\mathbf{\Delta}) := \mathrm{vec}(\mathbf{\Delta})^\top (\hat{\mathbf{H}} - \bar{\mathbf{H}}) \, \mathrm{vec}(\mathbf{\Delta}) = \mathrm{vec}(\mathbf{\Delta})^\top \left( \frac{1}{T_1} \sum_{t=1}^{T_1} \boldsymbol{x}_t \boldsymbol{x}_t^\top - \mathbb{E}[\boldsymbol{x}_t \boldsymbol{x}_t^\top] \right) \, \mathrm{vec}(\mathbf{\Delta})$$

$$= \frac{1}{T_1} \sum_{t=1}^{T_1} \left( (\boldsymbol{x}_t^\top \mathrm{vec}(\mathbf{\Delta}))^2 - \mathbb{E}[(\boldsymbol{x}_t^\top \mathrm{vec}(\mathbf{\Delta}))^2] \right)$$

$$= \frac{1}{T_1} \sum_{t=1}^{T_1} \left( \langle \mathbf{X}_t, \mathbf{\Delta} \rangle^2 - \mathbb{E}[\langle \mathbf{X}_t, \mathbf{\Delta} \rangle^2] \right)$$

$$= \frac{1}{T_1} \sum_{t=1}^{T_1} f_{\mathbf{\Delta}}(\mathbf{X}_t).$$

Consider the normalized set $\mathcal{C}_1 := \{ \mathbf{\Delta}' \in \mathcal{C} \mid \|\mathbf{\Delta}'\|_{\mathrm{nuc}} \leq 1 \}$. We aim to bound the supremum of the empirical process using Proposition 3:

$$Z_{\mathcal{C}_1} := \sup_{\mathbf{\Delta} \in \mathcal{C}_1} \left| \frac{1}{T_1} \sum_{t=1}^{T_1} f_{\mathbf{\Delta}}(\mathbf{X}_t) \right|,$$

for the function class $\mathcal{F}_{\mathcal{C}_1} = \{ f_{\mathbf{\Delta}} \mid \mathbf{\Delta} \in \mathcal{C}_1 \}$.

For the maximum variance, we have $\sup_{\mathbf{\Delta} \in \mathcal{C}_1} \mathrm{Var}(f_{\mathbf{\Delta}}(\mathbf{X}_t)) \leq \sup_{\mathbf{\Delta} \in \mathcal{C}_1} \mathbb{E}[\langle \mathbf{X}_t, \mathbf{\Delta} \rangle^4]$. In our setting, recall that each matrix $\mathbf{X}_i = (\mathbf{I} + \mathbf{L}_i)^{-1}$ in the arm set has eigenvalues in $(0, 1]$, which implies $\|\mathbf{X}_i\|_{\mathrm{op}} \leq 1$ and $\|\mathbf{X}_i\|_{\mathrm{F}} \leq \sqrt{|V|}$. Consider any $\mathbf{\Delta} \in \mathcal{C}_1$, and let the random variable $Z_t := \langle \mathbf{X}_t, \mathbf{\Delta} \rangle$ be drawn from the finite set $\{ \langle \mathbf{X}_1, \mathbf{\Delta} \rangle, \ldots, \langle \mathbf{X}_K, \mathbf{\Delta} \rangle \}$. Then, $|Z_t|$ is deterministically bounded as:

$$|Z_t| = |\langle \mathbf{X}_t, \mathbf{\Delta} \rangle| \leq \|\mathbf{X}_t\|_{\mathrm{op}} \|\mathbf{\Delta}\|_{\mathrm{nuc}} \leq \|\mathbf{\Delta}\|_{\mathrm{nuc}} \leq 1.$$

We use this uniform boundedness to control the fourth moment $\mathbb{E}[Z_t^4] = \mathbb{E}[Z_t^2 \cdot Z_t^2] \leq \|\mathbf{\Delta}\|_{\mathrm{nuc}}^4 \leq 1$. Thus $\sigma^2 = 1$. We need a uniform bound on $|f_{\mathbf{\Delta}}(\mathbf{X}_t)| \leq \sup_{\mathbf{\Delta} \in \mathcal{C}, t} |\langle \mathbf{X}_t, \mathbf{\Delta} \rangle^2|$. Again, using the property of the trace inner product: $|\langle \mathbf{X}_t, \mathbf{\Delta} \rangle| \leq \|\mathbf{X}_t\|_{\mathrm{op}} \|\mathbf{\Delta}\|_{\mathrm{nuc}}$. Then we have $\sup_{\mathbf{\Delta} \in \mathcal{C}, t} |f_{\mathbf{\Delta}}(\mathbf{X}_t)| \leq 2 \cdot \|\mathbf{\Delta}\|_{\mathrm{nuc}}^2 \leq 2$, meaning $b = 2$.

Next, we aim to bound

$$\mathbb{E}[Z_{\mathcal{C}_1}] = \mathbb{E} \sup_{\mathbf{\Delta} \in \mathcal{C}_1} \left| \frac{1}{T_1} \sum_{t=1}^{T_1} f_{\mathbf{\Delta}}(\mathbf{X}_t) \right|.$$

Let $\{\varepsilon_i\}_{i=1}^{T_1}$ be an i.i.d. Rademacher sequence. Then, we have

$$\mathbb{E} \left[ \sup_{\mathbf{\Delta} \in \mathcal{C}_1} \left| \frac{1}{T_1} \sum_{t=1}^{T_1} f_{\mathbf{\Delta}}(\mathbf{X}_t) \right| \right] \leq 2 \mathbb{E} \left[ \sup_{\mathbf{\Delta} \in \mathcal{C}_1} \left| \frac{1}{T_1} \varepsilon_t \sum_{t=1}^{T_1} \langle \mathbf{X}_t, \mathbf{\Delta} \rangle^2 \right| \right]$$

$$\leq 4 \mathbb{E} \left[ \sup_{\mathbf{\Delta} \in \mathcal{C}_1} \left| \frac{1}{T_1} \sum_{t=1}^{T_1} \varepsilon_t \langle \mathbf{X}_t, \mathbf{\Delta} \rangle \right| \right]$$

$$\leq 4 \|\mathbf{\Delta}\|_{\mathrm{nuc}} \frac{1}{T_1} \mathbb{E} \left[ \| \sum_{t=1}^{T_1} \varepsilon_t \mathbf{X}_t \|_{\mathrm{op}} \right]$$

where the first inequality is due to Lemma 1 (Proposition 4.11 of Wainwright (2019)), the second and third inequalities are due to the Talagrand contraction inequality and $|\langle \mathbf{X}_t, \mathbf{\Delta} \rangle| \leq \|\mathbf{\Delta}\|_{\mathrm{nuc}}$. By Proposition 4 with $v := \max\{ \| \sum_{t=1}^{T_1} \mathbf{X}_t \mathbf{X}_t^\top \|_{\mathrm{op}}, \, \| \sum_{t=1}^{T_1} \mathbf{X}_t^\top \mathbf{X}_t \|_{\mathrm{op}} \} \leq \sum_{t=1}^{T_1} \|\mathbf{X}_t\|_{\mathrm{op}}^2 \leq T_1$, we obtain

$$\mathbb{E} \Big\| \sum_{t=1}^{T_1} \varepsilon_t \mathbf{X}_t \Big\|_{\mathrm{op}} \leq \sqrt{2 \, v \, \log(2d)} \leq \sqrt{2T_1 \log(2d)}.$$

Using it, we obtain

$$\mathbb{E}\left[\sup_{\boldsymbol{\Delta}\in\mathcal{C}_1}\left|\frac{1}{T_1}\sum_{t=1}^{T_1}f_{\boldsymbol{\Delta}}(\mathbf{X}_t)\right|\right] \leq 4\,\|\boldsymbol{\Delta}\|_{\mathrm{nuc}}\,\frac{1}{T_1}\mathbb{E}\left[\|\sum_{t=1}^{T_1}\varepsilon_t\mathbf{X}_t\|_{\mathrm{op}}\right]$$

$$\leq 4\,\|\boldsymbol{\Delta}\|_{\mathrm{nuc}}\,\frac{1}{T_1}\sqrt{2T_1\log(2|V|)}$$

$$= 4\,\|\boldsymbol{\Delta}\|_{\mathrm{nuc}}\sqrt{\frac{2\log(2|V|)}{T_1}}.$$

Therefore,

$$\mathbb{E}[Z_{\mathcal{C}_1}] \leq 4\,\|\boldsymbol{\Delta}\|_{\mathrm{nuc}}\sqrt{\frac{2\log(2|V|)}{T_1}} \leq 4\sqrt{\frac{2\log(2|V|)}{T_1}} \quad (\forall\boldsymbol{\Delta}\in\mathcal{C}_1).$$

With the parameters $b = 2, \sigma^2 = 1$, Proposition 3 states that for any $t > 0$:

$$\mathbb{P}\left(Z_{\mathcal{C}_1} \geq 2\mathbb{E}[Z_{\mathcal{C}_1}] + t\right) \leq \exp\left(-\frac{T_1 t^2}{8\sigma^2 + 4bt}\right) = \exp\left(-\frac{T_1 t^2}{8 + 8t}\right).$$

Setting the right-hand side to $\delta$ implies that for universal constants $C_1, C_2 > 0$, we have $t \leq C_1\sqrt{\frac{\log(1/\delta)}{T_1}} + C_2\frac{\log(1/\delta)}{T_1}$. Thus, with probability at least $1 - \delta$:

$$Z_{\mathcal{C}_1} \leq 2\mathbb{E}[Z_{\mathcal{C}_1}] + t \leq 16\sqrt{\frac{2\log(2|V|)}{T_1}} + C_1\sqrt{\frac{\log(1/\delta)}{T_1}} + C_2\frac{\log(1/\delta)}{T_1}. \qquad (16)$$

Recall that the function $f_{\boldsymbol{\Delta}}$ is quadratic in $\boldsymbol{\Delta}$, meaning $f_{c\boldsymbol{\Delta}} = c^2 f_{\boldsymbol{\Delta}}$. This implies that the process $\nu(\boldsymbol{\Delta})$ follows the scaling rule $\nu(c\boldsymbol{\Delta}) = c^2\nu(\boldsymbol{\Delta})$. Consequently, we can write:

$$|\nu(\boldsymbol{\Delta})| = \left|\nu\left(\frac{\boldsymbol{\Delta}}{\|\boldsymbol{\Delta}\|_{\mathrm{nuc}}}\cdot\|\boldsymbol{\Delta}\|_{\mathrm{nuc}}\right)\right| = \|\boldsymbol{\Delta}\|_{\mathrm{nuc}}^2\left|\nu\left(\frac{\boldsymbol{\Delta}}{\|\boldsymbol{\Delta}\|_{\mathrm{nuc}}}\right)\right|.$$

Therefore, by Eq. (16), for any $\boldsymbol{\Delta}\in\mathcal{C}$, with probability at least $1 - \delta$:

$$|\nu(\boldsymbol{\Delta})| \leq \|\boldsymbol{\Delta}\|_{\mathrm{nuc}}^2\,Z_{\mathcal{C}_1} \leq C\,\|\boldsymbol{\Delta}\|_{\mathrm{nuc}}^2\left(\sqrt{\frac{\log(2|V|)}{T_1}} + \frac{\log(1/\delta)}{T_1}\right),$$

where we have absorbed all numerical constants and the less dominant square-root term into a single universal constant $C$. This completes the proof.

$\square$

## C  PROOF OF PROPOSITION 1

### C.1  AUXILIARY RESULTS

We state Proposition 10.6 in Wainwright (2019) when adapted for our exploration phase with length $T_1$ in Algorithm 1 for the rank-one case.

**Proposition 7** (Adaptation of Proposition 10.6 in Wainwright (2019))**.** *Suppose that the observation operator* $\Phi_{T_1}$ *satisfies the RSC condition in Assumption 2 with* $\kappa > 0$, *and* $\tau_{T_1}^2 \geq 0$:

$$\frac{1}{2T_1}\|\Phi_{T_1}(\boldsymbol{\Delta})\|_2^2 \geq \frac{\kappa}{2}\|\boldsymbol{\Delta}\|_F^2 - \tau_{T_1}^2\|\boldsymbol{\Delta}\|_{\mathrm{nuc}}^2 \quad (\forall\boldsymbol{\Delta}\in\mathcal{C}) \qquad (17)$$

*Then conditioned on the event* $\mathcal{G}(\lambda_{T_1}) := \left\{\left\|\frac{1}{T_1}\sum_{i=1}^{T_1}\eta_i\mathbf{X}_i\right\|_{\mathrm{op}} \leq \frac{\lambda_{T_1}}{2}\right\}$, *any nuclear-norm regularized least squares estimator in Eq. (5) satisfies the bound*

$$\|\widehat{\boldsymbol{\Theta}} - \boldsymbol{\Theta}^*\|_F^2 \ \leq\ \frac{9}{2}\frac{\lambda_{T_1}^2}{\kappa^2},$$

*valid for* $128\tau_{T_1}^2 \leq \kappa$.

Next, we introduce the following concentration inequality.

**Theorem C.1** (Matrix Gaussian Series Concentration (Tropp, 2012, Theorem 1.2)). *Let* $\{\mathbf{A}_k\}$ *be a finite sequence of fixed, self-adjoint matrices in* $\mathbb{R}^{|V|\times|V|}$, *and let* $\{\xi_k\}$ *be independent standard normal or Rademacher random variables. Then, for all* $t \geq 0$,

$$\mathbb{P}\left\{\lambda_{\max}\left(\sum_k \xi_k \mathbf{A}_k\right) \geq t\right\} \leq |V|\cdot\exp\left(-\frac{t^2}{2\sigma^2}\right), \quad \text{where} \quad \sigma^2 := \left\|\sum_k \mathbf{A}_k^2\right\|_{\text{op}}.$$

### C.2   PROOF OF PROPOSITION 1

*Proof.* Proposition 7 conditions the analysis on the event $\mathcal{G}(\lambda_{T_1}) := \left\{\left\|\frac{1}{T_1}\sum_{i=1}^{T_1}\eta_i\mathbf{X}_i\right\|_{\text{op}} \leq \frac{\lambda_{T_1}}{2}\right\}$, which bounds the empirical gradient at the truth in the dual norm. The noise term is small relative to the penalty, so the random cross term is dominated and the estimation error stays in a low-rank cone. We show that this event occurs with high probability.

We use a standard matrix concentration result from Theorem C.1 to the Gaussian series $\sum_{i=1}^{T_1}\eta_i\mathbf{A}_i$, where $\sigma^2 := \left\|\sum_{i=1}^{T_1}\mathbf{A}_i^2\right\|_{\text{op}}$ and $\eta_i \sim \mathcal{N}(0,1)$ are independent, we have that for all $t \geq 0$,

$$\mathbb{P}\left(\lambda_{\max}\left(\sum_{i=1}^{T_1}\eta_i\mathbf{A}_i\right) \geq t\right) \leq |V|\cdot\exp\left(-\frac{t^2}{2\sigma^2}\right). \tag{18}$$

Recall that for any matrix $\mathbf{A} \succeq 0$, the operator norm satisfies:

$$\|\mathbf{A}\|_{\text{op}} = \|\mathbf{A}\|_2 = \lambda_{\max}(\mathbf{A}),$$

where $\|\cdot\|_2$ is the spectral norm, and $\lambda_{\max}(\mathbf{A})$ is the largest eigenvalue of $\mathbf{A}$. These quantities coincide for symmetric matrices, since their singular values equal the absolute values of their eigenvalues. In our setting, each matrix $\mathbf{X}_i \in \mathbb{R}^{|V|\times|V|}$ is symmetric and defined as

$$\mathbf{X}_i = (\mathbf{I} + \mathbf{L}_i)^{-1},$$

where $\mathbf{L}_i \succeq 0$ is a positive semidefinite matrix (e.g., a graph Laplacian). Because the eigenvalues of $\mathbf{L}_i$ are nonnegative, the eigenvalues of $\mathbf{X}_i$ lie in the interval $(0, 1]$. In particular, we have:

$$\lambda_{\max}(\mathbf{X}_i) = \frac{1}{1 + \lambda_{\min}(\mathbf{L}_i)} \leq 1.$$

Therefore,

$$\|\mathbf{X}_i\|_{\text{op}} = \lambda_{\max}(\mathbf{X}_i) \leq 1 \quad \text{for all } \mathbf{X}_i \in \mathcal{X}.$$

Define the scaled matrices $\mathbf{A}_i := \frac{1}{T_1}\mathbf{X}_i$. Since

$$\left\|\sum_{i=1}^{T_1}\mathbf{X}_i^2\right\|_{\text{op}} \leq \sum_{i=1}^{T_1}\left\|\mathbf{X}_i^2\right\|_{\text{op}} = \sum_{i=1}^{T_1}\|\mathbf{X}_i\|_{\text{op}}^2 \leq \sum_{i=1}^{T_1}1 = T_1,$$

we have

$$\left\|\sum_{i=1}^{T_1}\mathbf{A}_i^2\right\|_{\text{op}} = \frac{1}{T_1{}^2}\left\|\sum_{i=1}^{T_1}\mathbf{X}_i^2\right\|_{\text{op}} \leq \frac{1}{T_1}.$$

Given the above bound $\left\|\sum_{i=1}^{T_1}\mathbf{A}_i^2\right\|_{\text{op}} \leq \frac{1}{T_1}$, setting $t \geq \sqrt{\frac{2\log(2|V|/\delta)}{T_1}}$ is sufficient to guarantee $t \geq \sqrt{2\left\|\sum_{i=1}^{T_1}\mathbf{A}_i^2\right\|_{\text{op}}\log\left(\frac{|V|}{\delta}\right)}$. This ensures the event in Eq. (18) occurs with a failure probability

of at most $\delta$, since

$$t^2 \geq 2 \left\| \sum_{i=1}^{T_1} \mathbf{A}_i^2 \right\|_{\text{op}} \log\left(\frac{|V|}{\delta}\right) \Leftrightarrow |V| \cdot \exp\left(-\frac{t^2}{2 \left\| \sum_{i=1}^{T_1} \mathbf{A}_i^2 \right\|_{\text{op}}}\right) \leq \delta.$$

By the union bound, the same result holds for the operator norm with a slightly adjusted constant, ensuring that with probability at least $1 - \delta$:

$$\left\| \frac{1}{T_1} \sum_{i=1}^{T_1} \eta_i \mathbf{X}_i \right\|_{\text{op}} \leq \sqrt{\frac{2 \log(2|V|/\delta)}{T_1}}.$$

To satisfy the condition of Proposition 7, we now choose our regularization parameter $\lambda_{T_1} = 2\sqrt{\frac{2 \log(2|V|/\delta)}{T_1}}$, this gives

$$\Pr[\mathcal{G}(\lambda_{T_1})] = \Pr\left[\left\{ \left\| \frac{1}{T_1} \sum_{i=1}^{T_1} \eta_i \mathbf{X}_i \right\|_{\text{op}} \leq \frac{\lambda_{T_1}}{2} \right\}\right] \geq 1 - \delta.$$

Finally, by Proposition 7, with probability at least $1 - \delta$

$$\|\widehat{\boldsymbol{\Theta}} - \boldsymbol{\Theta}^*\|_F^2 \leq \frac{9}{2} \frac{\lambda_{T_1}^2}{\kappa^2} = \frac{9}{2} \frac{\left(2\sqrt{2 \log(2|V|/\delta)/T_1}\right)^2}{\kappa^2} = \frac{36 \log(2|V|/\delta)}{\kappa^2 \, T_1}.$$

$\square$

# D PROOF OF THEOREM 4.1

## D.1 AUXILIARY RESULTS

We introduce the Davis–Kahan theorem, which relates the deviation between eigenspaces of symmetric matrices to perturbations in the matrix.

**Theorem D.1** (Davis–Kahan $\sin\theta$ Theorem (Yu et al., 2015, Theorem 1)). *Let $\boldsymbol{\Sigma}, \widehat{\boldsymbol{\Sigma}} \in \mathbb{R}^{p \times p}$ be symmetric matrices. Suppose $\mathbf{M} \in \mathbb{R}^{p \times d}$ and $\widehat{\mathbf{M}} \in \mathbb{R}^{p \times d}$ are matrices with orthonormal columns corresponding to eigenspaces of $\boldsymbol{\Sigma}$ and $\widehat{\boldsymbol{\Sigma}}$, respectively. Let $\delta_M$ denote the minimum eigengap between the eigenvalues corresponding to $\mathbf{M}$ and the rest. Then:*

$$\| \sin\theta(\widehat{\mathbf{M}}, \mathbf{M}) \|_{\text{F}} \leq \frac{\|\widehat{\boldsymbol{\Sigma}} - \boldsymbol{\Sigma}\|_{\text{F}}}{\delta_M}.$$

## D.2 PROOF OF THEOREM 4.1

**Armset rotation.** Recall that the ground truth matrix is $\boldsymbol{\Theta}^* = \boldsymbol{s}\boldsymbol{s}^\top$, where $\boldsymbol{s} \in \mathbb{R}^{|V|}$ is the vector of innate opinions. Let $\widehat{\boldsymbol{s}} \in \mathbb{R}^{|V|}$ denote the top eigenvector of the estimator $\widehat{\boldsymbol{\Theta}}$ in Stage 1. We complete $\widehat{\boldsymbol{s}}$ into an orthonormal basis of $\mathbb{R}^{|V|}$ by selecting an orthonormal matrix $\widehat{\mathbf{S}}_\perp \in \mathbb{R}^{|V| \times (|V|-1)}$ such that

$$\begin{bmatrix} \widehat{\boldsymbol{s}} & \widehat{\mathbf{S}}_\perp \end{bmatrix} \in \mathbb{R}^{|V| \times |V|} \quad \text{is an orthogonal matrix.}$$

That is, $\widehat{\mathbf{S}}_\perp$ spans the orthogonal complement of $\widehat{\boldsymbol{s}}$, satisfying $\widehat{\mathbf{S}}_\perp^\top \widehat{\boldsymbol{s}} = 0$ and $\widehat{\mathbf{S}}_\perp^\top \widehat{\mathbf{S}}_\perp = \mathbf{I}_{|V|-1}$.

We then rotate each arm $\mathbf{X} \in \mathcal{X}$ into the new orthonormal basis defined by $[\widehat{\boldsymbol{s}}, \widehat{\mathbf{S}}_\perp]$, resulting in:

$$\mathbf{X}' := \begin{bmatrix} \widehat{\boldsymbol{s}} & \widehat{\mathbf{S}}_\perp \end{bmatrix}^\top \mathbf{X} \begin{bmatrix} \widehat{\boldsymbol{s}} & \widehat{\mathbf{S}}_\perp \end{bmatrix} = \begin{bmatrix} \widehat{\boldsymbol{s}}^\top \mathbf{X} \widehat{\boldsymbol{s}} & \widehat{\boldsymbol{s}}^\top \mathbf{X} \widehat{\mathbf{S}}_\perp \\ \widehat{\mathbf{S}}_\perp^\top \mathbf{X} \widehat{\boldsymbol{s}} & \widehat{\mathbf{S}}_\perp^\top \mathbf{X} \widehat{\mathbf{S}}_\perp \end{bmatrix}.$$

Let $k := 2|V| - 1$ be the projected dimension. We extract the projected feature vector $\mathbf{x}_{\text{sub}} \in \mathbb{R}^k$ by collecting the first row and column of $\mathbf{X}'$:

$$
\mathbf{x}_{\text{sub}}(\mathbf{X}) := \begin{bmatrix} \mathbf{X}'_{1,1} \\ \mathbf{X}'_{2:|V|,1} \\ \mathbf{X}'_{1,2:|V|} \end{bmatrix} = \begin{bmatrix} \widehat{\boldsymbol{s}}^\top \mathbf{X} \widehat{\boldsymbol{s}} \\ \widehat{\mathbf{S}}_\perp^\top \mathbf{X} \widehat{\boldsymbol{s}} \\ \widehat{\boldsymbol{s}}^\top \mathbf{X} \widehat{\mathbf{S}}_\perp \end{bmatrix}.
$$

Similarly, define the projected signal vector $\theta_{\text{sub}}^* \in \mathbb{R}^k$ as:

$$
\theta_{\text{sub}}^* := \begin{bmatrix} \widehat{\boldsymbol{s}}^\top \boldsymbol{\Theta}^* \widehat{\boldsymbol{s}} \\ \widehat{\mathbf{S}}_\perp^\top \boldsymbol{\Theta}^* \widehat{\boldsymbol{s}} \\ \widehat{\boldsymbol{s}}^\top \boldsymbol{\Theta}^* \widehat{\mathbf{S}}_\perp \end{bmatrix}.
$$

We adapt the proof strategy of Theorem 4.3 in Kang et al. (2022) to our *OPD-Min-ESTR* algorithm under a rank-one signal assumption.

**The instantaneous regret in Stage 2.** Let $\mathbf{X}^* \in \arg\min_{\mathbf{X} \in \mathcal{X}} \langle \mathbf{X}, \boldsymbol{\Theta}^* \rangle$ be the optimal arm and $\mathbf{X}_t \in \mathcal{X}$ be the arm selected by the learner at time $t \in [T_2]$ during the State 2. We write $\boldsymbol{x}_{\text{sub}}^* := \boldsymbol{x}_{\text{sub}}(\mathbf{X}^*)$ and $\boldsymbol{x}_{t,\text{sub}} := \boldsymbol{x}_{\text{sub}}(\mathbf{X}_t)$. The instantaneous regret is:

$$
r_t := \langle \mathbf{X}^*, \boldsymbol{\Theta}^* \rangle - \langle \mathbf{X}_t, \boldsymbol{\Theta}^* \rangle.
$$

We decompose this regret as:

$$
r_t = \underbrace{\langle \mathbf{X}^*, \boldsymbol{\Theta}^* \rangle - \langle \boldsymbol{x}_{\text{sub}}^*, \boldsymbol{\theta}_{\text{sub}}^* \rangle}_{\text{(A) projection error for } \mathbf{X}^*} + \underbrace{\langle \boldsymbol{x}_{\text{sub}}^* - \boldsymbol{x}_{t,\text{sub}}, \boldsymbol{\theta}_{\text{sub}}^* \rangle}_{\text{(B) regret within subspace}} + \underbrace{\langle \boldsymbol{x}_{t,\text{sub}}, \boldsymbol{\theta}_{\text{sub}}^* \rangle - \langle \mathbf{X}_t, \boldsymbol{\Theta}^* \rangle}_{\text{(C) projection error for } \mathbf{X}_t}.
$$

Term (A) and (C) represent the approximation error due to the projection into the estimated subspace, while term (B) is the regret of the low-dimensional linear bandit problem.

We bound the projection errors, terms (A) and (C), denoted by $r_t^{\text{proj}} := (\langle \mathbf{X}^*, \boldsymbol{\Theta}^* \rangle - \langle \boldsymbol{x}_{\text{sub}}^*, \boldsymbol{\theta}_{\text{sub}}^* \rangle) + (\langle \boldsymbol{x}_{t,\text{sub}}, \boldsymbol{\theta}_{\text{sub}}^* \rangle - \langle \mathbf{X}_t, \boldsymbol{\Theta}^* \rangle)$

As the only component not captured in the subspace representation lies in the orthogonal complement $\widehat{\mathbf{S}}_\perp$, the projection error for $\mathbf{X}$ can be written solely in terms of the orthogonal complement subspace:

$$
\langle \mathbf{X}, \boldsymbol{\Theta}^* \rangle - \langle \boldsymbol{x}_{\text{sub}}(\mathbf{X}), \boldsymbol{\theta}_{\text{sub}}^* \rangle = \langle \widehat{\mathbf{S}}_\perp^\top \mathbf{X} \widehat{\mathbf{S}}_\perp, \ \widehat{\mathbf{S}}_\perp^\top \boldsymbol{\Theta}^* \widehat{\mathbf{S}}_\perp \rangle
$$

The total projection error can thus be expressed as:

$$
r_t^{\text{proj}} = \langle \widehat{\mathbf{S}}_\perp^\top \mathbf{X}^* \widehat{\mathbf{S}}_\perp, \widehat{\mathbf{S}}_\perp^\top \boldsymbol{\Theta}^* \widehat{\mathbf{S}}_\perp \rangle - \langle \widehat{\mathbf{S}}_\perp^\top \mathbf{X}_t \widehat{\mathbf{S}}_\perp, \widehat{\mathbf{S}}_\perp^\top \boldsymbol{\Theta}^* \widehat{\mathbf{S}}_\perp \rangle.
$$

Applying Cauchy-Schwarz inequality $|\langle A, B \rangle| \le \|A\|_F \|B\|_F$ for each term gives:

$$
|\langle \widehat{\mathbf{S}}_\perp^\top \mathbf{X}^* \widehat{\mathbf{S}}_\perp, \widehat{\mathbf{S}}_\perp^\top \boldsymbol{\Theta}^* \widehat{\mathbf{S}}_\perp \rangle| \le \|\widehat{\mathbf{S}}_\perp^\top \mathbf{X}^* \widehat{\mathbf{S}}_\perp\|_F \cdot \|\widehat{\mathbf{S}}_\perp^\top \boldsymbol{\Theta}^* \widehat{\mathbf{S}}_\perp\|_F,
$$

$$
|\langle \widehat{\mathbf{S}}_\perp^\top \mathbf{X}_t \widehat{\mathbf{S}}_\perp, \widehat{\mathbf{S}}_\perp^\top \boldsymbol{\Theta}^* \widehat{\mathbf{S}}_\perp \rangle| \le \|\widehat{\mathbf{S}}_\perp^\top \mathbf{X}_t \widehat{\mathbf{S}}_\perp\|_F \cdot \|\widehat{\mathbf{S}}_\perp^\top \boldsymbol{\Theta}^* \widehat{\mathbf{S}}_\perp\|_F.
$$

We write $\max_{\mathbf{X}} \|\mathbf{X}\|_F \le S_X$ for some constant $S_X > 0$. Since orthogonal projection cannot increase the Frobenius norm $\|\widehat{\mathbf{S}}_\perp^\top \mathbf{X} \widehat{\mathbf{S}}_\perp\|_F \le \|\mathbf{X}\|_F \le S_X$ for any $\mathbf{X}$, we have

$$
r_t^{\text{proj}} \le \left( \|\widehat{\mathbf{S}}_\perp^\top \mathbf{X}^* \widehat{\mathbf{S}}_\perp\|_F + \|\widehat{\mathbf{S}}_\perp^\top \mathbf{X}_t \widehat{\mathbf{S}}_\perp\|_F \right) \cdot \|\widehat{\mathbf{S}}_\perp^\top \boldsymbol{\Theta}^* \widehat{\mathbf{S}}_\perp\|_F \le 2S_X \cdot \|\widehat{\mathbf{S}}_\perp^\top \boldsymbol{\Theta}^* \widehat{\mathbf{S}}_\perp\|_F.
$$

Using $\boldsymbol{\Theta}^* = \boldsymbol{s} \boldsymbol{s}^\top$, we have:

$$
\|\widehat{\mathbf{S}}_\perp^\top \boldsymbol{\Theta}^* \widehat{\mathbf{S}}_\perp\|_F = \|(\widehat{\mathbf{S}}_\perp^\top \boldsymbol{s})(\boldsymbol{s}^\top \widehat{\mathbf{S}}_\perp)\|_F = \|\widehat{\mathbf{S}}_\perp^\top \boldsymbol{s}\|_2^2.
$$

The above analysis concludes that

$$
r_t^{\text{proj}} \le 2S_X \|\widehat{\mathbf{S}}_\perp^\top \boldsymbol{s}\|_2^2. \tag{19}
$$

Now, we aim to establish an explicit relation between $\|\widehat{\mathbf{S}}_\perp^\top s\|$ and the angle between the vectors $s$ and $\widehat{s}$, in order to apply the Davis–Kahan $\sin\theta$ theorem. To proceed, we decompose $s$ into a component aligned with $\widehat{s}$ and a residual orthogonal component:

$$s = \frac{\langle \widehat{s}, s \rangle}{\|\widehat{s}\|^2} \cdot \widehat{s} + r, \quad \text{where } r \perp \widehat{s}.$$

Projecting onto the orthogonal complement:

$$\widehat{\mathbf{S}}_\perp^\top s = \widehat{\mathbf{S}}_\perp^\top r \quad \Rightarrow \quad \|\widehat{\mathbf{S}}_\perp^\top s\| = \|r\|.$$

Using the Pythagorean theorem:

$$\|r\|^2 = \|s\|^2 - \left\| \frac{\langle \widehat{s}, s \rangle}{\|\widehat{s}\|} \right\|^2 = \|s\|^2 - \frac{\langle s, \widehat{s} \rangle^2}{\|\widehat{s}\|^2}.$$

Recalling that

$$\cos(\theta) = \frac{\langle s, \widehat{s} \rangle}{\|s\| \cdot \|\widehat{s}\|},$$

we obtain:

$$\|r\|^2 = \|s\|^2 (1 - \cos^2 \theta) = \|s\|^2 \cdot \sin^2 \theta.$$

Therefore, we conclude:

$$\|\widehat{\mathbf{S}}_\perp^\top s\| = \|s\| \cdot \sin(\theta).$$

In our setting, we apply Theorem D.1 with $\Sigma = \Theta^*$ and $\widehat{\Sigma} = \widehat{\Theta}$. Since $\Theta^* = ss^\top$ is rank-one, its top eigenspace is spanned by $s$, and the orthogonal complement corresponds to $\widehat{\mathbf{S}}_\perp$. Its largest eigenvalue is $\lambda_1(\Theta^*) = \lambda_{\max}(\Theta^*) = \|s\|_2^2$, and all other eigenvalues are zero. Therefore, the eigengap for the top eigenspace of the true matrix $\Theta^*$ is $\delta_M = \lambda_1(\Theta^*) - \lambda_2(\Theta^*) = \|s\|^2$.

Thus, we obtain:

$$\|\widehat{\mathbf{S}}_\perp^\top s\| = \|s\| \cdot \sin(\theta) \leq \|s\| \cdot \frac{\|\widehat{\Theta} - \Theta^*\|_F}{\|s\|^2} = \frac{\|\widehat{\Theta} - \Theta^*\|_F}{\|s\|}.$$

Substituting this inequality into Eq. (19) yields:

$$r_t^{\text{proj}} \leq 2 S_X \cdot \left( \frac{\|\widehat{\Theta} - \Theta^*\|_F}{\|s\|} \right)^2.$$

From Theorem 1, with probability at least $1 - \delta$,

$$\|\widehat{\Theta} - \Theta^*\|_F^2 \leq \frac{36 \log(2|V|/\delta)}{\kappa^2 T_1}.$$

Thus, the instantaneous projection error is bounded as:

$$r_t^{\text{proj}} \leq 72 S_X \cdot \frac{\log(2|V|/\delta)}{\|s\|^2 \kappa^2 T_1}. \tag{20}$$

**Final regret bound.** We now incorporate the remaining components of the regret.

During the initial exploration phase $t = 1, \ldots, T_1$, we incur the linear regret, so we conservatively bound the cumulative regret by applying Cauchy–Schwarz:

$$r_t = \langle \mathbf{X}^*, \Theta^* \rangle - \langle \mathbf{X}_t, \Theta^* \rangle \leq \|\mathbf{X}^* - \mathbf{X}_t\|_F \cdot \|\Theta^*\|_F \leq 2 S_X \cdot \|s\|^2.$$

Thus, the exploration cost is bounded as:

$$\sum_{t=1}^{T_1} r_t \leq 2 T_1 \cdot S_X \cdot \|s\|^2.$$

We denote the cumulative regret incurred by the linear bandit algorithm (e.g., OFUL (Abbasi-Yadkori et al., 2011)) is $R_T^{\text{sub}} = \tilde{\mathcal{O}}(k\sqrt{T})$, where $k = 2|V| - 1$ is the subspace dimension. Combining all terms, the total regret is:

$$
\begin{aligned}
R_T &= \sum_{t=1}^{T_1} r_t + \sum_{t=T_1+1}^{T} r_t \\
&= \sum_{t=1}^{T_1} r_t + \sum_{t=T_1+1}^{T} r_t^{\text{proj}} + R_T^{\text{sub}} \\
&\leq \underbrace{2T_1 \cdot S_X \cdot \|\boldsymbol{s}\|^2}_{\text{exploration cost}} + \underbrace{72 S_X \cdot \frac{\log(2|V|/\delta)}{\|\boldsymbol{s}\|^2 \kappa^2 \, T_1} \cdot T}_{\text{bias due to misalignment}} + \underbrace{c(k)\sqrt{T}}_{\text{in-subspace linear regret}} ,
\end{aligned}
\tag{21}
$$

where we used Eq. (20) and $c(k)$ is the constant term dependent on arm size $k$ in $R_T^{\text{sub}}$. When we set $T_1 := \Theta(\frac{1}{\|\boldsymbol{s}\|^2 \kappa} \sqrt{T \log(2|V|/\delta)})$, this gives:

$$
R_T = \mathcal{O}\left( \frac{S_X}{\kappa} \sqrt{T \log(|V|/\delta)} + c(k)\sqrt{T} \right)
$$

Since each matrix $\mathbf{X} = (\mathbf{I} + \mathbf{L})^{-1} \in \mathbb{R}^{|V| \times |V|}$ is symmetric positive semidefinite with eigenvalues in $(0, 1]$, we have:

$$
S_X^2 = \max_{\mathbf{X}} \|\mathbf{X}\|_F^2 = \max_{\mathbf{X}} \text{Tr}(\mathbf{X}^2) = \max_{\mathbf{X}} \sum_{i=1}^{|V|} \lambda_i^2(\mathbf{X}) \leq |V|.
$$

Finally, we obtain

$$
R_T = \tilde{\mathcal{O}}\left( \frac{1}{\kappa} \sqrt{T \cdot |V|} + |V|\sqrt{T} \right) = \tilde{\mathcal{O}}\left( \max\left\{ \frac{1}{\kappa}, \sqrt{|V|} \right\} \sqrt{|V| \cdot T} \right),
$$

where we used $c(k) = \tilde{\mathcal{O}}(k) = \tilde{\mathcal{O}}(|V|)$ as a standard regret bound for linear bandits. From the assumption, we have $\kappa = \kappa_{\min}(\mathcal{X})$, which completes the proof.

### D.3 PROOF OF COROLLARY 1

*Proof.* We have Eq. (21) with the same analysis in Theorem 4.1.

$$
R_T \leq \underbrace{2T_1 \cdot S_X \cdot \|\boldsymbol{s}\|^2}_{\text{exploration cost}} + \underbrace{72 S_X \cdot \frac{\log(2|V|/\delta)}{\|\boldsymbol{s}\|^2 \kappa^2 \, T_1} \cdot T}_{\text{bias due to misalignment}} + \underbrace{c(k)\sqrt{T}}_{\text{in-subspace linear regret}}.
$$

Set

$$
T_1 = \frac{6}{\ell_s \, \kappa} \sqrt{T \log \frac{2|V|}{\delta}} \qquad \text{with} \qquad \|\boldsymbol{s}\|^2 \geq \ell_s.
$$

Substituting into the above inequality gives

$$
R_T \leq 12 \frac{S_X}{\kappa} \left( \frac{\|\boldsymbol{s}\|^2}{\ell_s} + \frac{\ell_s}{\|\boldsymbol{s}\|^2} \right) \sqrt{T \log \frac{2|V|}{\delta}} + c(k)\sqrt{T} \leq \frac{24 \, S_X}{\kappa} \frac{\|\boldsymbol{s}\|^2}{\ell_s} \sqrt{T \log \frac{2|V|}{\delta}} + c(k)\sqrt{T}.
$$

Using $\|\boldsymbol{s}\|^2 \leq |V|$, $\ell_s = \Omega(|V|)$, $S_X \leq \sqrt{|V|}$ and $c(k) = \tilde{\mathcal{O}}(|V|)$, we obtain

$$
R_T = \tilde{\mathcal{O}}\left( \frac{|V|^{1/2}}{\kappa} \sqrt{T} \right).
$$

$\square$

# E  IMPLEMENTATION DETAILS

This section outlines the codebase and experimental settings used in our study. Additional notes on computational complexity are provided in Appendix F.1.

## E.1  CODE AND REPRODUCIBILITY

Code repository: `https://github.com/FedericoCinus/online-min-pol`.

The repository is organized around a main `src/` directory with modules for graph generation, subspace estimation, bandit optimization, and visualization. Each experimental setting can be reproduced with dedicated scripts, and all figures are automatically stored in the `figures/` directory. A `yaml` file specifying the `conda` environment is provided to ensure full reproducibility.

## E.2  EXPERIMENTAL SETTINGS

We summarize in Table 1 all parameters used in our experiments. The parameter grid is designed to study factors that directly influence opinion dynamics, putting less emphasis on standard bandit hyperparameters. In the main text we report results for representative values near the middle of each range, while additional experiments with extreme values are deferred to this appendix.

**Data.** Real-world graphs are taken from the `NetworkX` library, and all synthetic graphs can be reproduced using the provided code. No additional preprocessing was applied.

Table 1: Overview of experimental parameters, grouped by component: *Graph models*, *Opinions*, *Arms*, *Noise environment*, *General setting*, *Stage 1 (subspace estimation)*, and *Stage 2 (OFUL optimization)*.

| Description | Symbol / Values |
|---|---|
| *Graph models* | |
| Number of nodes | $|V| \in \{8, 16, 32, 34, 64, 77, 256, 1024\}$ (synthetic + real) |
| Stochastic Block Model | Two-community, homophilic: $|V_1| \approx 0.75|V|, |V_2| = |V| - |V_1|$ |
| | Intra-community $p = 0.5$, inter-community $p = 0.07$ |
| Erdős–Rényi Model | Edge probability $p = 0.2$ |
| *Opinions* | |
| Opinion vector | $\boldsymbol{s} \sim \mathrm{Unif}([-1, 1]^{|V|})$, mean-centered |
| *Arms* | |
| Number of arms | $|\mathcal{X}| \in \{10, 100, 1000\}$ |
| Single arm generation | $\mathbf{X}_t$: $|V|$ random rank-one updates of $\mathbf{L}$, weight $\sim \mathrm{Unif}[0.5, 1.5]$ |
| *Noise environment* | |
| Noise variance | $\sigma_\eta \in \{0.01, 0.1, 1.0\}$ |
| *General setting* | |
| Confidence parameter | $\delta = 0.001$ |
| Time horizon | $T = 10{,}000$ |
| Stage 1 duration | $T_1 = \sqrt{T}$ |
| Stage 2 duration | $T_2 = T - T_1$ |
| *Stage 1: Subspace estimation* | |
| Nuclear-norm weight | $\lambda_{\mathrm{nuc}} = \frac{2}{\sqrt{T_1}}\sqrt{\log(2d/10^{-2})}$ |
| *Stage 2: OFUL optimization* | |
| Regularization | $\lambda = 0.1$ |
| Arm norm bound | $L_x = |V|$ (conservative upper bound) |
| Parameter norm bound | $\|\mathbf{s}\|^2 = |V|$ (conservative upper bound) |

# F    ALGORITHMIC DETAILS

In this section, we analyze the computational complexity of each stage and show how OPDMin maps to the linear bandit framework of `OFUL`, verifying the necessary assumptions.

## F.1    COMPUTATIONAL COMPLEXITY

We analyze the time and memory complexity of our Algorithm 1, separating the *theoretical* requirements of each stage from the properties of our current implementation and possible optimizations. Let $|V|$ denote the number of nodes, $T$ the horizon, $T_1$ the Stage-1 budget, and $\mathcal{X}$ the arm set.

**Stage-1: Subspace Estimation.**    We solve a nuclear-norm regularized problem using proximal gradient descent. Each iteration involves: (i) computing a gradient over $T_1$ sampled arms, costing $\mathcal{O}(T_1|V|^2)$, and (ii) applying a singular value thresholding (SVT) step, which requires a full SVD of a $|V| \times |V|$ matrix at cost $\mathcal{O}(|V|^3)$. With $K$ iterations, the total complexity is

$$\mathcal{O}\big(K|V|^3 + KT_1|V|^2\big).$$

This complexity is inherent to the method, though in practice the SVD can be replaced with randomized or power methods that approximate the top singular directions, reducing the cost closer to $\mathcal{O}(|V|^2 \log|V|)$ per iteration.

**Stage-1 Projection.**    After estimating $\hat{s}$, each arm $\mathbf{X} \in \mathbb{R}^{|V| \times |V|}$ must be projected onto a basis aligned with $\hat{s}$ and reduced to a $(2|V| - 1)$-dimensional feature vector. Naively forming the rotated matrix would cost $\mathcal{O}(|V|^3)$ per arm, but by computing only the necessary bilinear forms our low-memory implementation reduces this to $\mathcal{O}(|V|^2)$ operations per arm, for a total time of $\mathcal{O}(|\mathcal{X}||V|^2)$. Moreover, we never materialize full $|V| \times |V|$ matrices in memory: instead, each arm is stored directly in its reduced form, requiring only $\mathcal{O}(|\mathcal{X}||V|)$ storage overall.

**Stage-2: OFUL in Reduced Dimension.**    Let $p = 2|V| - 1$ denote the reduced dimension. Each round of OFUL requires solving for $\mathbf{A}^{-1}x$ for all arms at cost $\mathcal{O}(|\mathcal{X}|p^2)$, plus updates to the design matrix $\mathbf{A} \in \mathbb{R}^{p \times p}$ and vector $\mathbf{b} \in \mathbb{R}^p$, costing $\mathcal{O}(p^2)$. Thus, the overall complexity of Stage-2 is

$$O\big(T|\mathcal{X}|p^2\big).$$

**Comparison with Standard `OFUL`.**    Running OFUL directly in the full $|V|^2$-dimensional space would require $\mathcal{O}(|\mathcal{X}||V|^4)$ operations per round and $\mathcal{O}(|V|^4)$ memory, which is prohibitive even for moderate $|V|$. By contrast, our reduction to $p = 2|V| - 1$ dimensions lowers the per-round cost to $\mathcal{O}(|\mathcal{X}||V|^2)$ and the memory usage to $\mathcal{O}(|V|^2)$. This reduction preserves theoretical guarantees while yielding orders-of-magnitude improvements in both time and memory efficiency.

## F.2    ADAPTING OPDMIN TO OFUL

The `OFUL` algorithm (Abbasi-Yadkori et al., 2011) is a standard approach for linear bandits that maintains confidence sets around the unknown parameter and plays optimistically. To apply it in our OPDMin setting, we rewrite the quadratic loss $\mathbf{s}^\top \mathbf{X} \mathbf{s}$ as a linear form. Let $\mathbf{\Theta}^* = \mathbf{s}\mathbf{s}^\top$, $\theta = \mathrm{vec}(\mathbf{\Theta}^*)$, and $\mathbf{x} = \mathrm{vec}(\mathbf{X})$. Then the loss is $\mathbf{s}^\top \mathbf{X} \mathbf{s} = \langle \mathbf{X}, \mathbf{\Theta}^* \rangle = \langle \mathbf{x}, \theta^\star \rangle$. At each round $t$, the learner selects an arm $\mathbf{x}_t$, observes the noisy loss

$$y_t = \langle \mathbf{x}_t, \theta^\star \rangle + \eta_t,$$

and updates its estimate of the unknown parameter $\theta^\star$.

**Norm bounds on actions.**    For $\mathbf{x} = \mathrm{vec}(\mathbf{X})$, we have $\|\mathbf{x}\|_2 = \|\mathbf{X}\|_F$. Since $\mathbf{X} = (\mathbf{I} + \mathbf{L})^{-1}$ with $\mathbf{L}$ a graph Laplacian, all eigenvalues lie in $(0, 1]$. Thus $\|\mathbf{X}\|_2 \leq 1$ and $\|\mathbf{X}\|_F \leq \sqrt{|V|}$, giving $\|\mathbf{x}\|_2 \leq \sqrt{|V|}$.

**Norm bounds on the unknown parameter.**    The parameter is $\theta^\star = \mathrm{vec}(\mathbf{s}\mathbf{s}^\top)$, with $\|\theta^\star\|_2 = \|\mathbf{s}\mathbf{s}^\top\|_F = \|\mathbf{s}\|_2^2$. Since each $s_i \in [-1, 1]$, we obtain $\|\theta^\star\|_2 \leq |V|$.

Therefore, the OFUL assumptions are satisfied with parameter norm bound $S = |V|$ and action norm bound $L_x = \sqrt{|V|}$.

# G    ADDITIONAL EXPERIMENTS

In this section of the appendix, we provide extended experimental results that complement the main text. We evaluate the robustness of our method under different network models, real-world datasets, large-scale settings, and sensitivity analyses.

## G.1    EMPIRICAL VALIDATION OF THE RSC PARAMETERS

The Restricted Strong Convexity (RSC) condition (Definition 4) requires that the quadratic form in Eq. (12) is uniformly bounded below across all admissible directions $\boldsymbol{\Delta} \in \mathcal{C}$, up to a tolerance term. Formally, it balances a positive curvature term with constant $\kappa$ against an additional tolerance proportional to $\|\boldsymbol{\Delta}\|_{\mathrm{nuc}}^2$.

For empirical validation, we focus only on the curvature component. Following standard RSC-style analysis, we empirically compute the observation operator as

$$\hat{\kappa} = \min_{\boldsymbol{\Delta} \in \mathcal{C},\, \|\boldsymbol{\Delta}\|_F = 1} \frac{1}{T_1} \|\Phi_{T_1}(\boldsymbol{\Delta})\|_2^2,$$

which approximates the normalized curvature across admissible directions. Although this proxy ignores the tolerance term in the formal definition, it still provides a conservative diagnostic for whether the sampled arms induce sufficient curvature to recover the low-rank structure. The unit Frobenius constraint is without loss of generality, since the quadratic form is homogeneous and depends only on the direction of $\boldsymbol{\Delta}$.

Computing this minimum exactly is intractable due to the high-dimensional, non-convex search space. We therefore approximate $\hat{\kappa}$ via a projected gradient descent (PGD) heuristic. At each iteration, we normalize $\boldsymbol{\Delta}$ to satisfy $|\boldsymbol{\Delta}|_F = 1$, enforce cone membership, and project to rank-2 (motivated by the fact that the difference of two rank-1 matrices has rank at most 2). Multiple random restarts are used to mitigate local minima.

The resulting $\hat{\kappa}$ is only an approximate estimate of the curvature proxy defined above. Thus, $\hat{\kappa}$ cannot be interpreted as a rigorous lower bound on the theoretical RSC constant from Definition 4. Instead, it should be viewed as a practical diagnostic: when $\hat{\kappa}$ is reasonably large, this provides empirical evidence that the sampled arms induce sufficient curvature to make the problem well-conditioned.

**Experimental setup.**   We compare two graph families and two arm-generation regimes. For Erdős–Rényi (ER) graphs we use edge probability $p = 0.2$. For Stochastic Block Model (SBM) graphs we consider a homophilic two-community structure with 75% of the nodes in the first community and 25% in the second, intra-community probability $p = 0.5$, and inter-community probability $p = 0.07$. Arms are generated either (i) *Local*, by applying $2|V|$ random edge edits to a fixed base Laplacian, or (ii) *Diverse*, by independently sampling fresh random Laplacians (ER for ER graphs, homophilic SBM for SBM graphs). In all cases we fix $|\mathcal{X}| = 100$. Reported values are averaged over 25 trials. Very small estimates in the *Local* regime reflect nearly collinear arms, which induce weak curvature.

Table 2: Empirical values of $\hat{\kappa}$ (mean $\pm$ std) across graph families and arm regimes. See main text for detailed description of graph models and arm generation procedures.

| Graph | Arm regime | $|V| = 32$ | $|V| = 128$ | $|V| = 1024$ |
|-------|-----------|-----------|------------|-------------|
| ER | Diverse | 0.393 ($\pm$0.037) | 0.410 ($\pm$0.045) | 0.499 ($\pm$0.039) |
| ER | Local | $(1.68 \pm 0.40) \times 10^{-5}$ | $(1.49 \pm 0.19) \times 10^{-7}$ | $(2.21 \pm 0.00) \times 10^{-7}$ |
| SBM | Diverse | 0.386 ($\pm$0.040) | 0.476 ($\pm$0.017) | 0.462 ($\pm$0.026) |
| SBM | Local | $(2.97 \pm 0.43) \times 10^{-6}$ | $(8.05 \pm 0.08) \times 10^{-7}$ | $(1.05 \pm 0.00) \times 10^{-7}$ |

## G.2    SCALABILITY

To empirically assess scalability, we evaluate our algorithm on Erdős–Rényi graphs with increasing node sizes, up to $|V| = 1024$. Figure 2 reports the wall-clock time (averaged over trials) as a function

of the network size. The results show near-polynomial growth consistent with our complexity analysis, confirming that the algorithm remains practical for graphs of moderate size.

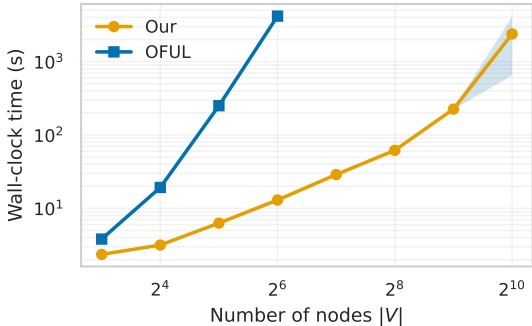

Figure 2: Wall-clock time of OPD-Min as a function of the number of nodes $|V|$ on Erdős–Rényi graphs. Shaded regions indicate standard deviation across trials.

### G.3    EXPERIMENTS UNDER POLARIZED OPINION DISTRIBUTIONS

To study the impact of stronger polarization, we generate innate opinions using a bimodal distribution that concentrates mass near the extremes $-1$ and $+1$. Specifically, for opinions $s_i \sim \mathrm{Unif}([-1, 1])$, we apply the transformation $s_i \mapsto \mathrm{sign}(s_i)\,|s_i|^{1/3}$, which amplifies values closer to $\pm 1$ while compressing those near 0.

Figure 3 shows the resulting cumulative regret curves on Erdős–Rényi graphs. We observe behavior similar to the uniform setting, with nearly identical regret curves. However, the polarized case achieves faster reduction of polarization and lower absolute regret, suggesting that interventions are easier to identify and exploit when opinions are more extreme.

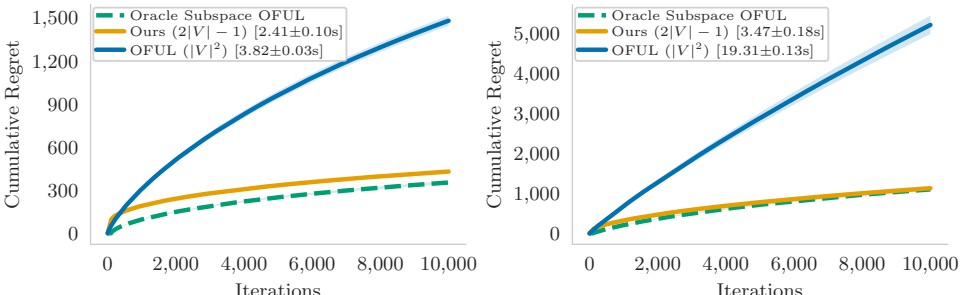

Figure 3: Cumulative regret on Erdős–Rényi graphs with polarized innate opinions (pol = 3). Runtime (mean $\pm$ std) over 100 repetitions is reported in the legend. Edge probability is $p = 0.2$.

### G.4    RESULTS ON REAL-WORLD NETWORKS

We evaluate our algorithm on three widely studied real-world social networks: (i) the Florentine families network Breiger & Pattison (1986), representing marriage ties among prominent families in Renaissance Florence, (ii) the Davis Southern women network Davis et al. (1941), a bipartite affiliation network connecting women to the social events they attended, (iii) the Karate club network Zachary (1977), capturing friendships among members of a university karate club, and (iv) the Les Misérables network Knuth (1993), describing co-occurrences of characters in Victor Hugo's novel. All four are standard benchmark graphs available in NetworkX[1]. Figure 4 illustrates the cumulative regret over 10,000 iterations on real-world networks, evaluated under different noise levels and action set sizes. The results highlight the robustness and efficacy of our algorithm across heterogeneous real graph topologies.

---

[1]https://networkx.org/documentation/stable/auto_examples/index.html

For all real-data experiments, we generate candidate interventions using the same procedure as in our synthetic settings: each arm applies $2|V|$ random rank-one edge updates to the base Laplacian, with additive weight offsets sampled uniformly from $(0.5, 1.5)$.

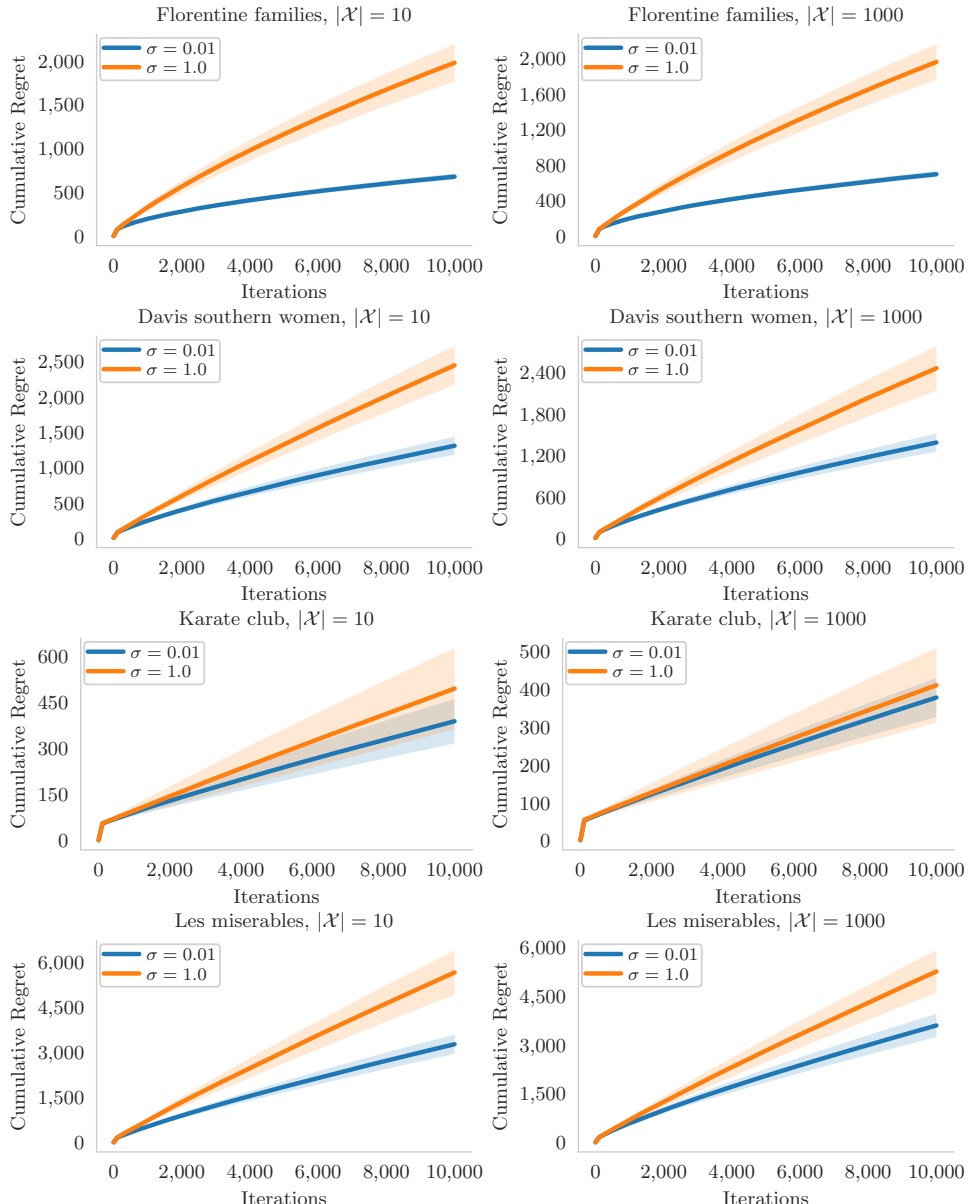

Figure 4: Cumulative regret over 10,000 iterations on four benchmark social networks (Florentine families, Davis southern women, Karate club, and Les Misérables), under two noise levels ($\sigma = 0.01$ and $\sigma = 1.0$). The left column corresponds to action set size $|\mathcal{X}| = 10$, while the right column corresponds to $|\mathcal{X}| = 1000$. Each curve shows the mean regret across runs, with shaded regions indicating 95% confidence intervals.

### G.5 SENSITIVITY ANALYSIS

We conduct a series of sensitivity analyses to study the robustness of our algorithm with respect to key environment parameters.

Figure 4 reports the cumulative regret over 10,000 iterations on real networks under varying *noise levels* ($\sigma = 0.01$ and $\sigma = 1.0$) and *action set sizes* ($|\mathcal{X}| = 10$ on the left, $|\mathcal{X}| = 1,000$ on the right).

Across all datasets, the cumulative regret of our method grows predictably with the horizon, with lower regret observed for smaller action sets and lower noise levels.

### G.6 COMPARISON WITH THE OFFLINE BASELINE

To empirically verify that online algorithms can outperform the offline SDP baseline of Chaitanya et al. (2024)—which, in our discrete setting, reduces to evaluating the objective on all arms and selecting the minimizer (Eq. 10)—we run two short experiments on small graphs where exact evaluation is feasible. We consider: (i) a stochastic block model with homophilic links, $|V| = 16$, $T = 250$, $T_1 = 50$, $\sigma = 10^{-4}$, $|\mathcal{X}| = 100$, and 500 trials; and (ii) the same setting with $|V| = 32$. For each method, we plot the *minimum polarization + disagreement* achieved so far as a function of the number of iterations (i.e., interventions). The curve for Chaitanya et al. (2024) is flat, since this offline method does not incorporate online feedback.

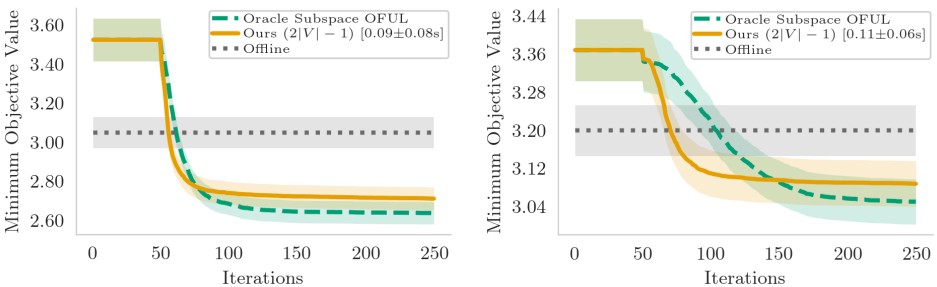

Figure 5: Minimum polarization + disagreement over 250 iterations (interventions) for our method, the oracle subspace OFUL, and the offline baseline. The offline baseline is flat. We also report the mean final objective values: for $|V| = 32$, ORACLE = 3.05 and OURS = 3.09; for $|V| = 16$, ORACLE = 2.64 and OURS = 2.71.

In both settings, the two online algorithms (ours and the oracle subspace OFUL) quickly drop below the offline optimum and continue to improve as additional samples are collected, demonstrating the benefit of sequential feedback. At the end of each run, we also report the mean objective of the final selected arm. These results confirm that, in this regime, online exploration yields strictly better interventions than the best offline solution.

