# OpenReview forum: "Online Minimization of Polarization and Disagreement via Low-Rank Matrix Bandits"
_ICLR.cc/2026/Conference — ICLR 2026 Poster_

### Official Review · Reviewer_jpz5 · 2025-11-01

**Soundness:** 3
**Presentation:** 4
**Contribution:** 2
**Rating:** 4
**Confidence:** 3

**Summary:**

The paper tackles minimizing polarization plus disagreement in the Friedkin–Johnsen model with online bandit feedback, observing one noisy scalar per intervention. It runs two phases: a low rank trace regression estimates a rank one matrix and its top direction to rotate and compress arm features to linear in the number of nodes, then a standard linear bandit such as OFUL operates on the reduced features; the authors prove phase one recovery and an overall regret that scales with the square root of time and network size, and validate empirically.

**Strengths:**

1/ The subject is highly relevant and important.

2/ The approach is clearly presented with an exemplary writing.

3/ The authors cleverly adapt classical bandit strategies to their specific setting.

**Weaknesses:**

Summary

While the paper has the merit of carefully deriving the technical aspects and adapts classical bandit concepts (a preconditioning phase followed by OFUL), it does not clearly justify the online formulation over prior offline formalisms (or a BAI formalisms) and omits state-of-the-art baselines in the experiments.

Major concerns

1/ Please compare with additional baselines (e.g., Chaitanya et al., 2024) and with the subspace Oracle in a sample-based evaluation. Using the same arm set and sample budget, plot the minimum polarization plus disagreement obtained versus the number of observed intervention outcome pairs (learning-curve style). For your method and the Oracle also report the mean polarization plus disagreement of the best arm selected after T = number of samples.

2/ Please justify the online regret minimization (vs. offline periodic updates and a BAI setting for example). Explain why interventions are sequential and interim loss matters a real case, and add a brief disscussion on the comparison to batched/BAI baseline in this aspect.

Minor comments

1/ Consider restricting the regret plot to the first 2,000 iterations and use a logarithmic x-axis/y-axis to better expose the early hinge between Phase-1 and Phase-2.

2/ There are numerous typos, for example: "which which we term", "problem reduces to a low-rank matrix bandits", "existing analyzes", etc

3/ In the supplementary material, expand the ethics/impacts discussion (potential misuse, beneficiaries vs. risks, etc). Also add a plain-language note on the meaning in real life of the mean-centered innate opinions assumption.

Grading explanation

The paper is clear and technically strong, yet key SOTA baselines are missing (and the motivation for the setting is not clear).

**Questions:**

1/ Why choosing a regret setting and not BAI setting in your case?

2/ Could you detail how the candidate arms are generated for the real-data experiments?

**Details Of Ethics Concerns:**

The paper proposes an outcome-agnostic intervention that can steer collective opinions, posing risks of suppressing dissent or critical views ("Manufacturing Consent"); I’m not qualified to assess this and recommend an expert ethics/broader-impacts review.

---

> ### Author Response · Authors · 2025-11-20
>
> We really appreciate your careful reading and invaluable feedback. All changes in the paper are marked in blue for your convenience.
>
> > Major 1: Baseline
>
> We have added subsection G.6 in the additional experiments that includes two short controlled experiments specifically designed to compare online methods against the offline baseline of Chaitanya et al. (2024). The new plot uses the number of samples (interventions) on the x-axis and the minimum polarization + disagreement achieved so far on the y-axis. It includes our method, the subspace-oracle variant, and the offline baseline. By definition, the curve for Chaitanya et al. (2024) is flat because this method does not use online feedback. We also report the mean final objective value of the arm selected by our method and by the oracle. The results consistently show that both online algorithms surpass the offline baseline after only a few iterations, reinforcing our main empirical findings and the inherent advantage of sequential feedback. From our perspective, this experiment confirms an expected behavior, but we are happy to keep it in the paper if the reviewers find it useful.

---

> ### Author Response · Authors · 2025-11-20
>
> > Major 2: Learning task
>
> We have added a short discussion at the end of Section 2 (Background), motivating our choice of an online regret-minimization framework, clarifying that it better reflects realistic sequential-intervention settings compared to batched or BAI formulations. See Q1 for more details.

---

> ### Author Response · Authors · 2025-11-20
>
> > Minor 1
>
> Although a log scale could highlight the Phase-1/Phase-2 transition, we keep the standard linear-scale plots for consistency with prior low-rank bandit and general bandit literature.

---

> ### Author Response · Authors · 2025-11-20
>
> > Minor 2
>
> Thank you for pointing these out. We have corrected the mentioned typos and performed a full proofreading pass to fix similar issues throughout the paper.

---

> ### Author Response · Authors · 2025-11-20
>
> > Minor 3
>
> We thank the reviewer for the helpful suggestion. We have included an Ethics and Impact discussion, highlighting potential misuse, beneficiaries versus risks, and noting that our study is theoretical and that only platform operators—not arbitrary malicious actors—have the access necessary to implement the types of interventions considered.
>
>
> Regarding the mean-centered assumption, we agree that a plain-language explanation is beneficial. We have added a concise justification in the paragraph where assumption (1) is introduced (Sec. 2.2). The revised text clarifies that mean-centering does not restrict generality but simply removes the global offset of opinions, which does not affect polarization, disagreement, or the equilibrium of the FJ model.

---

> ### Author Response · Authors · 2025-11-20
>
> > Question 1: Learning task
>
> In realistic scenarios, a recommender system pays a (social) cost each time it deploys an intervention, since every recommendation immediately affects users and system welfare. This naturally aligns with cumulative regret minimization, which accounts for these interim costs. In contrast, BAI assumes that exploration is cost-free until a final decision is made, which does not reflect how real platforms operate. Although BAI and batched formulations would be interesting future directions, especially from a theoretical perspective, establishing the online regret framework is, in our view, the necessary first step.

---

> ### Author Response · Authors · 2025-11-20
>
> > Question 2: Arm set generation
>
> We use the same arm-generation procedure as in our synthetic experiments. Each arm is created by applying 2|V| random rank-one edge updates to the original real-data Laplacian, with additive weight offsets sampled uniformly from (0.5, 1.5). We have added a clarifying sentence in the experimental section.

---

> ### Comment · Reviewer_jpz5 · 2025-11-21
> **Reviewer Response**
>
> Thank you for your precise response and for your time.
>
> So in Chaitanya et al. (2024), it seems that the upper bound is loose (p. 2: "However, the tightness of such an upper bound to the optimum remains unclear.").
>
> It did not occur to me that both disagreement and polarization can be understood as variance-like statistics, so offsets do not affect them.
>
> I would only add that, historically, influence attempts tend to come less from "arbitrary malicious actors" and more from well-known public figures or institutions (companies or governments) aiming to steer opinion or reduce disagreement; however, your ethical impact statement already seems sufficiently developed.
>
> My main concern was the (unjustified) introduction of an online setting without a clear performance comparison to existing methods, and your reply addressed this well.
>
> I thank the authors, and I raise my score to 8.

---

### Official Review · Reviewer_zipZ · 2025-11-02

**Soundness:** 3
**Presentation:** 4
**Contribution:** 3
**Rating:** 8
**Confidence:** 3

**Summary:**

This paper studies an online learning problem involving the Friedkin-Johnsen (FJ) opinion dynamics model. The model describes a discrete-time dynamical system over the expressed opinions of users connected by an interaction graph, which also depends on some innate opinions of the finitely many users, which initializes the system. The system is known to converge to an equilibrium (fixed-point), at which one can measure polarization (variance of the equilibrium opinion distribution) and disagreement (essentially some notion of smoothness of the equilibrium distribution as given by its quadratic form under the graph Laplacian). The authors study the fundamental question of whether, without knowledge of the initial, innate opinion distribution of the users, one can choose interaction graph structures to minimize polarization and disagreement. This is formulated as an online learning problem where, at each time step, the user chooses an interaction graph structure and receives bandit feedback on an objective function encoding the polarization and disagreement of the resulting FJ dynamics at equilibrium. The problem is motivated by social media platforms implementing interventions (via changes to user interaction structure) to minimize polarization and disagreement.

For this problem, the authors obtain a sublinear regret bound scaling like O(|V| \sqrt{T}) (where |V| is the number of users) using an explore-then-commit approach: first, using some uniform exploration phase to learn a low-rank subspace in the interaction graph space, and then to run a no-regret linear bandit algorithm on the set of actions in the learned, lower-dimensional representation. The result of the two-phase approach is an improved linear dependence on |V| in the final regret bound, as opposed to a quadratic dependence if one naively runs a standard linear bandit algorithm from the initial round. One of the core technical novelties of the work is the design of the initial subspace estimation phase, which is specialized to the particular discrete structure of the problem setting and departs from prior works on dimensionality reduction in matrix bandits.

The authors additionally show the effectiveness of the proposed method experimentally.

**Strengths:**

Overall, the paper is very clearly written and presented, and the new subspace estimation technique used in the main algorithm seems novel.

**Weaknesses:**

Several assumptions made in the paper could be discussed further:
* One assumption is that the intervention space (action space) is comprised of a fixed set of K admissible graph Laplacians, but it is not discussed how the regret of the algorithm scales with K (in particular if the action set contains all graph Laplacians).
* A second assumption is that the user observes an estimate of the polarization/disagreement objective at equilibrium under the FJ dynamics, but it is not discussed how fast the FJ dynamics actually converges to this equilibrium.

Note that the main body of the paper also exceeds 9 pages by several lines (but after communication with a PC representative, this is OK).

**Questions:**

Regarding the assumptions mentioned under Weaknesses:
* Are you assuming the intervention space is a priori finite and fixed due to some exogenous problem constraints (e.g., a platform can only choose certain network structures given privacy/connection constraints between users)? In principle, this set could be exponentially large in |V| (since each graph Laplacian corresponds to a different adjacency matrix). How does the final regret bound of Theorem 4.1 depend on the size of the intervention space K?
* Is it known how quickly the FJ dynamics converges to equilibrium (c.f., the comment regarding asymptotic convergence in L126-L127)? In other words, at what timescale with respect to the FJ dynamics does the outer OPD-Min-ESTR algorithm operate, since it is assumed that the learner receives bandit feedback on the objective function for the dynamics at equilibrium.

---

> ### Author Response · Authors · 2025-11-20
>
> We really appreciate your careful reading and invaluable feedback.
>
> > W1 / Q1: Arm set
>
> Yes, we assume the intervention space is a priori finite and fixed. From a modeling standpoint, a finite intervention set is realistic: in practice, a platform typically has access to only a limited catalogue of possible actions (e.g., recommending a curated set of new connections, injecting specific content, or applying moderation interventions), each corresponding to a predefined modification of user exposure or connectivity. Thus, the discrete action-space assumption reflects practical constraints while enabling efficient and theoretically grounded learning.
>
> A dynamic setting, where the intervention set changes at each time $t$, is a very interesting direction. We believe an extension is likely possible, but it would require a new analysis of the RSC conditions and the subspace estimation bounds in Phase 1. We expect that Phase 2 could be easily adapted to this setting.
>
> Regarding the size $K$ of the intervention space: Phase 2 essentially utilizes a standard linear bandit algorithm, which involves quadratic maximization over an ellipsoid confidence region. For computational simplicity, we assume the size of the intervention space, $K$, is within a range where this optimization is computationally feasible. However, to answer your specific question, the final regret bound in Theorem 4.1 is $K$-independent ($K$ does not appear in the bound).

---

> ### Author Response · Authors · 2025-11-20
>
> > W2 / Q2: FJ convergence
>
> Convergence is exponential with rate $\rho(D^{-1}A) < 1$ (where $\rho$ is the spectral radius). This follows from the standard FJ update $x(t+1) = D^{-1}(Ax(t) + s)$, whose error satisfies $e(t) = (D^{-1}A)^{t} e(0)$. Since the powers of $D^{-1}A$ decay at the rate $\rho(D^{-1}A)^{t}$, only a small number of iterations are needed to get close to equilibrium in practice. Thus, the FJ dynamics settle much faster than the outer OPD-Min-ESTR learning loop, and it is appropriate to treat the learner’s feedback as the equilibrium value of the objective. We added a clarifying sentence in Section 2.1 in response to this helpful comment. All changes in the paper are marked in blue for your convenience.

---

> ### Comment · Reviewer_zipZ · 2025-11-26
>
> Thanks to the authors for their replies. I remain positive about the paper and will maintain my score.

---

### Official Review · Reviewer_M4wL · 2025-11-09

**Soundness:** 4
**Presentation:** 4
**Contribution:** 3
**Rating:** 8
**Confidence:** 4

**Summary:**

This paper studies the problem of minimizing polarization and disagreement under the *Friedkin–Johnsen (FJ) opinion dynamics model* when there is incomplete information about the innate opinions. By defining actions as constructing different forest matrices $X = (I + L)^{-1}$, where $L$ is the Laplacian matrix of the graph, the authors formulate the minimization of polarization and disagreement as a multi-armed bandit problem. By sequentially intervening on the network graph, the algorithm aims to minimize the cumulative regret—the sum of expressed polarization and disagreement—over a finite horizon $T$.

The proposed method consists of two stages. Stage 1 performs uniform sampling over the action set to learn about the opinion subspace. Stage 2, using the noisy loss observations collected in Stage 1, performs dimensionality reduction and maps the linear bandit problem onto the reduced subspace. The proposed algorithm achieves a regret bound of  $\tilde{O}(|V|\sqrt{T})$,
where $|V|$ denotes the number of nodes in the graph.

The paper further compares the proposed algorithm with two baselines:  (1) applying OFUL directly on the full high dimensional space $R^{\mid V\mid ^2}$; (2) the oracle case where the true subspace is known, serving as a lower bound. By learning the subspace first, the proposed algorithm achieves substantially lower regret and faster convergence compared to performing OFUL on the full space.

**Strengths:**

This paper advances the study of polarization and disagreement minimization in the following ways:  (1) it assumes no prior knowledge about the innate opinion information; (2) it reformulates the minimization of polarization and disagreement under the FJ model within a multi-armed bandit framework.  The technical contribution of this paper lies in adapting previous MAB algorithms, which typically assume a continuous action space (e.g., Gaussian random matrices), to a setting where the action space is discrete, highly structured, and induced by the graph Laplacian. The paper provides a novel theoretical analysis of the regret bound that used the Restricted Strong Convexity (RSC) condition for the defined action set. The authors show that the RSC condition holds when performing uniform sampling over their specific action set. Furthermore, in the experimental section, the proposed algorithm’s performance closely matches the empirical lower bound corresponding to the oracle case, which has access to the true subspace $\Theta^*$.

I have skimmed through the proofs for the subspace reduction and regret analysis, and I do not see any obvious issues with them. However, since I am not very familiar with the RSC condition, I will refrain from commenting further on that aspect.

Overall, this paper is clearly written, and the notations are well defined and consistent.

**Weaknesses:**

1. Although it might be a common assumption in the FJ opinion dynamics model, this paper assumes that the innate opinion $s$ is static over time, and only the expressed opinion $z_t$ evolves. I am somewhat skeptical about this assumption, as innate opinions could also be influenced by the expressed opinions of neighbors. However, since this work explicitly follows the assumptions of the standard FJ model, this limitation may not significantly undermine the overall validity of the paper.

2. The rationale for why the action space is discrete is not clearly explained, nor is it entirely clear whether this assumption is reasonable within the given problem setting. (See Question 1.)

**Questions:**

1. It is not very clear why the action space is assumed to be discrete in the problem setup. By definition, the action is given as  $X_w = (I + L_w)^{-1}$, where $L_w = D - A_w$ is the Laplacian matrix of a graph. Since $A_w$ is the adjacency matrix of the graph $G$ with edge weights $w$, and these weights could form a continuous but constrained vector, it is not immediately obvious why the action space cannot be treated as continuous.

2. I would recommend revising the definition in line 287 to clarify that the subscript $t$ refers to $t \in T_1$ and that $[\cdot]_t$ denotes an entry of the vector. This notation was somehow unclear upon first reading.

---

> ### Author Response · Authors · 2025-11-20
>
> We really appreciate your careful reading and invaluable feedback.
>
>
> > W1: Assumption
>
> The distinction between a static innate opinion and an evolving expressed opinion is a core feature of the FJ framework. The model is designed precisely to separate an individual’s stable predisposition (the innate component) from the malleable, socially influenced part of their opinion. While different terminology could be adopted, this two-layer structure captures the idea that individuals may adjust what they express without fundamentally changing their underlying predisposition at the timescale considered. As the reviewer notes, our work adheres to this well-established modeling choice, so this assumption does not undermine our results and remains fully consistent with the standard FJ literature.

---

> ### Author Response · Authors · 2025-11-20
>
> > W2 / Q1: Arm set
>
> We thank the reviewer for raising this point. In our formulation, the action space is taken to be discrete for both computational and modeling reasons. From the algorithmic perspective, Phase 2 in the low-rank matrix bandits literature reduces to a standard linear bandit problem, where each action corresponds to selecting one intervention \Delta_k.
> In the context of the standard linear bandit setting (e.g., in algorithms like OFUL), selecting the action that maximizes the uncertainty (i.e., the width of the confidence interval) corresponds to a quadratic maximization problem over the ellipsoidal confidence region defined by the empirical covariance matrix. Optimizing over a continuous space of admissible Laplacian perturbations is computationally expensive and generally not tractable at scale. By restricting the action set to a finite collection, we ensure that this step remains efficient, while the regret bound of the algorithm remains independent of $K$.
> From a modeling standpoint, a finite intervention set is also realistic: in practice, a platform typically has access to only a limited catalogue of possible actions (e.g., recommending a curated set of new connections, injecting specific content, or applying moderation interventions), each corresponding to a predefined modification of user exposure or connectivity. Thus, the discrete action-space assumption reflects practical constraints while enabling efficient and theoretically grounded learning.

---

> > ### Comment · Reviewer_M4wL · 2025-11-25
> >
> > I thank the reviewer for addressing my question on the arm set selection. All my questions has been answered, and I think it is a good paper.

---

> ### Author Response · Authors · 2025-11-20
>
> > Q2: Notation
>
> Thank you for the suggestion — we have revised those lines accordingly. All changes in the paper are marked in blue for your convenience.

---

### Author Response · Authors · 2025-12-01
**Summary for the New AC**

We thank all reviewers for their careful evaluations and constructive feedback. Since scores were reverted, we briefly summarize how the rebuttal and discussion resolved their concerns.


**Clarifications on assumptions:**
Reviewers asked about the discrete arm set and the modeling assumptions in the Friedkin–Johnsen dynamics. We provided detailed explanations consistent with standard practice and computational constraints. Reviewers confirmed these points were fully addressed.


**Offline baseline comparison:**
In response to a reviewer request, we added an experiment comparing our online methods with the offline baseline of Chaitanya et al. (2024). As expected, the offline method—which does not learn over time—is quickly outperformed by our online algorithms after only a few iterations. All additions are marked in blue in the revised version.


**Final reviewer consensus:**
After discussion, all reviewers stated that their concerns were resolved and converged on the same positive score (8).

---

### Meta-Review · Area_Chair_JWAC · 2026-01-08

**Summary:**

This paper studies the online minimization of polarization and disagreement in the Friedkin–Johnsen opinion dynamics model under incomplete information. The authors consider a sequential setting in which innate opinions are unknown and only a scalar feedback of polarization and disagreement is observed after each intervention. The problem is formulated as a regret minimization task, and the paper proposes a two-stage low-rank matrix bandit algorithm that first estimates a low-dimensional subspace and then runs a linear bandit method in the reduced space. The authors provide regret guarantees scaling sublinearly in time and linearly in the number of nodes, and empirical results demonstrate improvements over a full-space linear bandit baseline.

Overall, reviewers agreed that this is a solid theory-driven work with clear analysis and supportive experiments. The main concerns raised by reviewers focused on the motivation for the online formulation, the discreteness and scope of the intervention space, the relationship to prior offline or best-arm identification approaches, and the adequacy of experimental baselines. One reviewer also noted modeling assumptions in the Friedkin–Johnsen framework and potential ethical considerations.

Overall, there are general agreement on the technical contributions of the work, and the concerns primarily are surrounding the justifications for the modeling choices.

**Reviewer Concerns:**

Most of the concerns are requests for more discussion on the modeling choices.  The authors have also provided reasonable reasons for each of them during the rebuttal.

**Reviewer Scores:**

Three reviewers gave initial scores of 8, and I expect their final scores to remain at 8.

One reviewer gave an initial score of 4 and indicated raising the score to 8 in the follow-up discussion. Even without considering that discussion, the authors provided reasonable responses to the main concerns regarding the motivation for the online setting and the inclusion of additional baselines, which would likely have increased the score to at least 6.

Overall, I believe this paper would have reached a positive consensus among the reviewers.

---

### Decision · Program_Chairs · 2026-01-26

Accept (Poster)